# Learning, Solving and Optimizing PDEs with TENSORGALERKIN: an efficient high-performance Galerkin assembly algorithm

**Shizheng Wen** [* 1]  **Mingyuan Chi** [*]  **Tianwei Yu** [1]  **Ben Moseley** [2]  **Mike Yan Michelis** [1]  **Pu Ren** [3]  **Hao Sun** [4]  **Siddhartha Mishra** [1]

Project page: https://www.tensor-mesh.com/

## Abstract

We present a unified algorithmic framework for the numerical solution, constrained optimization, and physics-informed learning of PDEs with a variational structure. Our framework is based on a Galerkin discretization of the underlying variational forms, and its high efficiency stems from a highly-optimized and GPU-compliant TENSOR-GALERKIN framework for linear system assembly (stiffness matrices and load vectors). TENSOR-GALERKIN operates by tensorizing element-wise operations within a Python-level Map stage and then performs global reduction with a sparse matrix multiplication that performs message passing on the mesh-induced sparsity graph. The Map and Reduce stages are co-designed inside PyTorch's autograd so that the assembly graph contains $O(1)$ nodes regardless of how the number of elements and local DoFs scale. We validate this $O(1)$-graph property by deploying TENSOR-GALERKIN downstream as i) a highly-efficient numerical PDEs solver, ii) an end-to-end differentiable framework for PDE-constrained optimization, and iii) a physics-informed operator learning algorithm for PDEs. With multiple benchmarks, including 2D and 3D elliptic, parabolic, and hyperbolic PDEs on unstructured meshes, we demonstrate that the proposed framework provides significant computational efficiency and accuracy gains over a variety of baselines in all the targeted downstream applications.

---

[*]Equal contribution  [1]ETH Zurich, Switzerland [2]Imperial College London, UK [3]Northeastern University, USA [4]Renmin University of China, China. Correspondence to: Shizheng Wen <shizheng.wen@sam.math.ethz.ch>.

*Proceedings of the $43^{rd}$ International Conference on Machine Learning*, Seoul, South Korea. PMLR 306, 2026. Copyright 2026 by the author(s).

## 1. Introduction

Partial Differential Equations (PDEs) mathematically model a very large variety of physical phenomena across a vast range of spatio-temporal scales (Evans, 2022). In the absence of analytical solution formulas, numerical methods are widely used to (approximately) *solve* PDEs (Quarteroni & Valli, 1994). Among them, the *finite element method* (FEM) stands out for its generality regarding complex domain geometries, robustness, mathematical guaranties, and performance (Brenner & Scott, 2010). However, numerical methods such as FEM can be computationally expensive, especially for so-called *many query* problems such as inverse design, PDE-constrained optimization, and uncertainty quantification (Quarteroni et al., 2015), necessitating the design of new paradigms for solving PDEs.

One such recent framework corresponds to *learning* the solution *operator* of PDEs from data (Mishra & Townsend, 2024). These data-driven ML approaches often fall under the rubric of *operator learning* (Chen & Chen, 1995; Kovachki et al., 2023; Bartolucci et al., 2023). A whole spectrum of operator learning algorithms, based on convolutions (Li et al., 2020; Raonic et al., 2023), GNNs (Pfaff et al., 2021; Fortunato et al., 2022; Mousavi et al., 2025), and transformers (Herde et al., 2024; Wen et al., 2026; Alkin et al., 2024; Wu et al., 2024; Hao et al., 2023), are successful at learning solution operators of PDEs if sufficient amounts of training data is available. However, these methods perform poorly in the low data regime or when evaluated in out-of-distribution settings (Herde et al., 2024).

Motivated by these limitations, *physics-informed machine learning* has emerged as an alternative to purely data-driven operator learning. Physics-informed neural networks (PINNs) (Lagaris et al., 1998; Raissi et al., 2019) enforce PDE constraints by minimizing strong-form residuals within neural network ansatz spaces, but are known to suffer from fundamental numerical and computational challenges (De Ryck & Mishra, 2024). In particular, strong-form formulations impose restrictive regularity requirements and lead to severely ill-conditioned optimization problems,

partially motivating variational alternatives such as Deep Ritz (E & Yu, 2018), VPINNs (Kharazmi et al., 2019), and weak PINNs (De Ryck et al., 2022). More broadly, PINN-based methods act as single-instance solvers and rely heavily on *automatic differentiation* (AD) to compute high-order spatial and temporal derivatives pointwise. This results in deep computational graphs, large memory footprints, and substantial runtime overheads, which are further exacerbated by element-wise quadrature loops, especially on large or unstructured meshes (De Ryck & Mishra, 2024; Wang & Perdikaris, 2021; Li et al., 2024; Bischof et al., 2025).

These issues with physics-informed machine learning provide the rationale for this paper, where our main goal is to *propose an efficient algorithmic paradigm for solving PDEs as well as learning them in a physics-informed manner*. To this end, we present a unified framework for solving, learning, and optimizing a wide class of PDEs that possess an intrinsic variational structure, expressed in terms of bilinear forms and (semi-)linear functionals. Motivated by FEM, we develop a Galerkin-based solving and learning framework for PDEs, and identify that the *key bottleneck* for efficient implementation—not only for our framework but also for general FEM and physics-informed methods—lies in the *assembly of stiffness matrices and load vectors*. While assembly is trivial in compiled languages (e.g., C++ or Fortran), integrating standard FEM assembly into Python-based automatic differentiation frameworks such as PyTorch introduces critical system-level bottlenecks that are often overlooked. Classical assembly procedures iterate over elements, and looping over $10^5$–$10^7$ elements in Python incurs substantial interpreter overhead and GPU underutilization. Even when element-wise operations are vectorized, loops over local basis indices typically remain, leading to highly fragmented computation graphs in AD frameworks. This fragmentation prevents effective kernel fusion and results in severe autograd overhead during backpropagation. Existing approaches only partially mitigate these issues: CPU-based libraries (e.g., scikit-fem (Gustafsson & McBain, 2020)) suffer from I/O bottlenecks, JIT-compiled frameworks (e.g., JAX-FEM (Xue et al., 2023)) incur re-compilation overheads for dynamic meshes, and ad-hoc PyTorch implementations often rely on nondeterministic atomic operations that degrade reproducibility.

To address these efficiency bottlenecks, our main contribution in this article is to present TENSORGALERKIN, an efficient high-performance framework that resolves these bottlenecks by reformulating Galerkin assembly as a strictly tensorized Map-Reduce operation. The Map stage evaluates the physics of the problem (in the form of the PDE variational form) in terms of dense tensor contractions, fusing the computation of all local stiffness matrices or load vectors into a *single GPU kernel*. On the other hand, the Reduce stage handles domain topology by replacing scatter-

add loops with a precomputed routing. Global assembly is then executed as a deterministic sparse matrix multiplication (SpMM). Consequently, this paradigm reduces computation to just a few monolithic nodes, minimizing Python overhead and maximizing instruction throughput. Crucially, routing both stages through PyTorch's autograd compresses the backward graph from $O(Ek^2)$ nodes (with $E$ elements and $k$ local degrees of freedom) down to $O(1)$, independent of mesh size, which eliminates the fragmentation overhead identified above.

With this highly efficient assembly paradigm at hand, and to validate the $O(1)$-graph property in practice, we employ TENSORGALERKIN for i) *solving* PDEs numerically with a PyTorch-native, highly optimized, GPU compatible FEM solver that we call TENSORMESH, which can efficiently handle dynamic meshes while significantly outperforming legacy CPU FEM solvers; ii) *learning* solution operators of PDEs in a data-free, physics-informed manner with our proposed TENSORPILS framework, without requiring automatic differentiation to compute spatial derivatives, as these are evaluated with *analytical shape gradients*; and iii) *optimizing* systems modeled by PDEs, for instance, in inverse design, as our entire TENSOROPT pipeline is end-to-end differentiable, and the gradients required for optimization are evaluated efficiently. For all three aspects, we evaluate our proposed paradigm on a set of challenging experiments and show that our approach is significantly superior to the baselines, both in terms of efficiency and accuracy.

## 2. Methods

**Problem Formulation.** For simplicity of exposition, we consider a time-independent *linear* PDE here to present our method. Extensions to time-dependent semi-linear PDEs are postponed to **SM A.1**. Let $\Omega \subset \mathbb{R}^d$ be the domain of interest. On it, we assume that the underlying PDE is defined in terms of its *variational structure* (Evans, 2022), where we seek a solution $u \in V$ such that

$$\mathsf{a}_\rho(u, v) = \ell_\rho(v) \quad \forall v \in W, \tag{1}$$

where $V$ and $W$ are suitable Banach spaces; the bilinear $\mathsf{a}_\rho(\cdot, \cdot) : V \times W \to \mathbb{R}$ models the underlying physics; the linear form $\ell_\rho(\cdot) : W \to \mathbb{R}$ represents external forcing. Moreover, $\rho \in \Pi$ (with $\Pi$ being another suitable Banach space) represents spatially-varying coefficients and forces that serve as *inputs* to the PDE. Concrete examples of the quantities occurring in Eq. (1) are provided in **SM A.2**. The solution operator $\mathsf{S} : \Pi \mapsto V$, corresponding to the PDE (1), is then given by $\mathsf{S}(\rho) = u$. The *operator learning task* (Bartolucci et al., 2023) amounts to finding an approximation of $\mathsf{S}_{\#\mu}$, given data drawn from an *input distribution* $\mu \in \mathrm{Prob}(\Pi)$.

**Neural Galerkin Discretization.** The key idea underpinning Galerkin (and wider finite element methods (Brenner & Scott, 2010)) is to approximate the infinite-dimensional spaces $V, W$ in Eq. (1) with *suitable finite-dimensional subspaces*. In this paper, we focus on the Galerkin method by considering the same approximation subspaces for both $V$ and $W$. To this end, we set $N > 1$ and define basis functions $\{\phi_i\}_{i=1}^N$. We realize an (approximate) solution of Eq. (1) in terms of the ansatz:

$$u_h^\theta(\mathbf{x}) = \sum_{j=1}^N U_j^\theta(\rho)\phi_j(\mathbf{x}) \quad \forall \mathbf{x} \in \Omega. \tag{2}$$

Here, $U^\theta = \{U_j^\theta\}_{j=1}^N$ is a family of functions parametrized by $\theta \in \Theta \subset \mathbb{R}^p$, which map the input functions $\rho$ into the coefficients $U_j^\theta$ of $u_h^\theta$. A particularly effective parametrization follows from setting $U^\theta$ as (deep) neural networks.

**Numerical PDE Solvers.** For a fixed $\bar{\rho} \in \Pi$ and setting the input distribution as $\mu = \delta_{\bar{\rho}}$, the operator learning task reduces to simply finding $S(\bar{\rho})$. Traditionally, this has been performed by the Galerkin method. In this case, we set $U_j^\theta \equiv U_j$ for all $\theta$ and $j$ and suppress the dependence on $\bar{\rho}$ for notational simplicity. The ansatz (2) is now given by $\sum_j U_j\phi_j$. Substituting it into the variational problem (1) and testing against $\phi_i$ reduces the variational problem to the *Matrix Equation*:

$$KU = F, \quad \text{where } K_{ij} := \mathsf{a}(\phi_j, \phi_i), \quad F_i := \ell(\phi_i). \tag{3}$$

Here, $\mathsf{a} = \mathsf{a}_{\bar{\rho}}$ and $\ell = \ell_{\bar{\rho}}$ are the corresponding bilinear and linear forms, respectively. In matrix equation (3), matrix $K$ is the *Galerkin stiffness matrix*, and $F$ the *load vector*.

**Physics-informed Operator Learning and Neural PDE Solvers.** Numerical PDE solvers are tailored for the *single query* case where the inputs (coefficients, sources) are just single instances of $\rho \equiv \bar{\rho}$ in Eq. (1). However, for operator learning, the input $\rho$ is sampled from a general data distribution $\mu \in \mathrm{Prob}(\Pi)$. In this case, a numerical PDE solver, such as the Galerkin solver defined above, will need to solve the matrix equation (3) *individually* for every sample $\rho$ at inference, leading to a very high computational cost. Moreover, this method does not *amortize* information that can be learned during an (offline) training stage for online inference.

However, the neural Galerkin method can be readily applied to this setting by considering the general ansatz (2). The following parameterized functions $U^\theta$ in the ansatz (1) can be calculated as the minimizer of the *Residual*.

$$\theta^* = \arg\min_{\theta \in \Theta} \mathcal{L}(\theta), \ \mathcal{L}(\theta) := \|K(\rho)U^\theta(\rho) - F(\rho)\|^2. \tag{4}$$

Here, $K_{ij}(\rho) := \mathsf{a}_\rho(\phi_j, \phi_i)$, $F_i(\rho) := \ell_\rho(\phi_i)$, and $\|\cdot\|$ is a vector-norm that can be further modulated by a mass

(preconditioner) matrix. Variants of gradient descent can be used to (approximately) solve the minimization problem (4) and find the coefficients $U^\theta$ of the ansatz (2). During the (offline) training phase, the minimization problem (4) is solved for $M$ inputs $\rho$ sampled from the distribution $\mu$. During the (online) inference phase, unseen values of $\rho$ are provided as input to the parameterized functions $U^{\theta^*}$, and their outputs form the (approximate) solution (2). In general, our approach is *data-free* as no labeled data pairs $(\rho, S(\rho))$ are needed. Thus, it constitutes *physics-informed operator learning*. However, this approach can easily ingest labeled data by adding a data-fidelity loss term to the residual (4), resulting in a hybrid data-driven and physics-informed operator learning approach. In the special case of *single query problems* with $\mu = \delta_{\bar{\rho}}$, our neural Galerkin operator learning algorithm boils down to a *neural PDE solver* such as PINNs, VPINNs, weak PINNs, or Deep Ritz.

**Algorithmic Realization and Bottlenecks.** It is clear that whether we consider solving PDEs (3) or physics-informed operator learning (4), an algorithmic realization of our framework consists of two steps, namely, i) the Galerkin stiffness matrix $K$ and the load vector $F$ need to be computed, and ii) the resulting linear system in (3) or the optimization problem in (4) needs to be solved. While we rely on classical iterative solvers (Saad, 2003) to solve linear systems in Eq. (3) and on (stochastic) gradient descent algorithms such as ADAM and L-BFGS for solving the minimization problem (4), we focus our attention here on computing the Galerkin stiffness matrix efficiently. The load vector computation follows an analogous workflow and is omitted in the following discussion for brevity.

We start with the standard **Scatter-Add** procedure for computing global stiffness matrices in a FEM context (Brenner & Scott, 2010). We consider a general unstructured partition of the domain $\Omega = \cup_{m=1}^E e_m$, with non-overlapping elements $e_m$. Let $k$ denote the number of local degrees of freedom (DoFs) associated with each element. The *local Galerkin matrix* $K_e \in \mathbb{R}^{k \times k}$ is computed via quadrature:

$$(K_e)_{ab} = \mathsf{a}_\rho(\phi_b^{(e)}, \phi_a^{(e)}) \approx \sum_{q=1}^Q \hat{w}_q(\dots) \tag{5}$$

for $a, b \in \{1, \dots, k\}$. Here, $\{\phi_a^{(e)}\}_{a=1}^k$ denotes the basis functions restricted to element $e$, and $\{\hat{w}_q\}_{q=1}^Q$ represents the quadrature weights. Let $g_e : \{1, \dots, k\} \to \{1, \dots, N\}$ be the mapping from local DoF to global DoF indices. The classical assembly process accumulates local contributions into the global system via a *scatter-add* reduction:

$$K_{g_e(a),g_e(b)} \leftarrow K_{g_e(a),g_e(b)} + (K_e)_{ab} \tag{6}$$

for $e \in \{1, \dots, E\}$. As described in detail in the introduction, this scatter-add framework is computationally inefficient in the context of AD paradigms due to the inherent

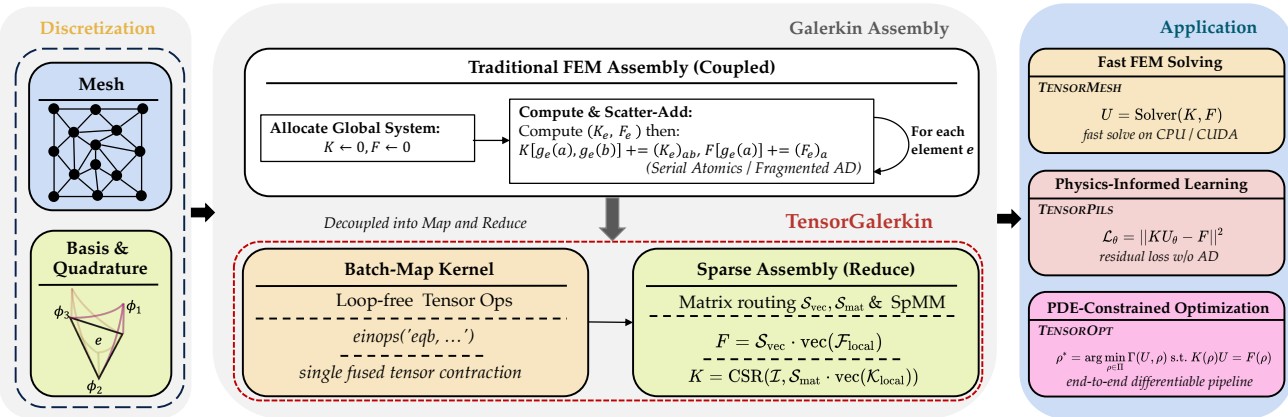

Figure 1. Overview of TENSORGALERKIN. Stage I (Batch-Map) computes element-wise operators via a fully tensorized einsum kernel; Stage II (Sparse-Reduce) assembles global sparse values via routing matrices and a single SpMM. For comparison, the white box illustrates traditional FEM assembly via per-element loops and scatter-add (atomics) into the global system. The same assembly engine powers TENSORMESH, TENSORPILS, and TENSOROPT.

looping over a large number of elements, the resulting fragmentation of the computational graph, and the severe overhead during backpropagation. These bottlenecks will be addressed in our novel framework that we describe below.

**The TENSORGALERKIN Framework.** It is clear from the structure of the scatter-add paradigm that its inherent sequential accumulation is responsible for significant overheads within modern AD frameworks. Hence, within the TENSORGALERKIN framework, we will replace this sequential procedure by reformulating the Galerkin assembly process into a strictly tensorized *Map-Reduce* paradigm. This architecture will decouple local physics computation (Map) from global topological aggregation (Reduce), allowing each stage to be executed as a monolithic GPU kernel.

*Stage I: Batch-Map (Fully Tensorized Physics)*

The Map stage computes local element contributions $\mathcal{K}_{\text{local}} = \{K_e\}_{e=1}^{E}$ in parallel. Unlike standard implementations that iterate over elements, we lift the element index to a batch dimension and perform dense tensor contractions.

Partition $\Omega \subset \mathbb{R}^d$ using a mesh of $E$ elements $\{e_m\}_{m=1}^{E}$. Each element $e$ is endowed with $k$ DoFs (associated with, e.g., nodes and edges). The geometric information is encoded by the batched coordinates tensor $\mathcal{X} \in \mathbb{R}^{E \times k \times d}$. Let $\{(\hat{w}_q, \hat{\mathbf{x}}_q)\}_{q=1}^{Q}$ be the weights and nodes of a $Q$-point quadrature rule defined on the reference element $\hat{e}$. For a generic bilinear form $a_\rho(\cdot, \cdot)$, the local Galerkin matrices $\{K_e\}_{e=1}^{E}$ are computed by contracting the basis gradients with the physical coefficients over the quadrature dimension. To clarify, let $\mathcal{G} \in \mathbb{R}^{E \times Q \times k \times d}$ denote the batched gradients of the basis functions transformed to the physical domain, and let $\mathcal{C} \in \mathbb{R}^{E \times Q \times \cdots}$ denote the batched point evaluation

of the coefficients. The assembly of all $E$ local matrices is expressed as a single tensor contraction:

$$(\mathcal{K}_{\text{local}})_{eab} = \sum_{q=1}^{Q} \hat{w}_q |\det(\mathcal{J}_{eq})| \cdot \mathcal{F}(\mathcal{G}_{eqa}, \mathcal{G}_{eqb}, \mathcal{C}_{eq}). \quad (7)$$

Here, $\mathcal{J}_{eq}$ denotes the Jacobian of the geometric mapping $\Phi_e$ that transforms the reference element $\hat{e}$ to the physical element $e$ (Brenner & Scott, 2010) evaluated on $\hat{\mathbf{x}}_q$; $\mathcal{F}$ encodes the bilinear form $a_\rho$ in terms of the coefficient $\rho$ and the gradients of the basis functions. A concrete example illustrating Eq. (7) is provided in **SM A.2**. In PyTorch, this computation is implemented via `torch.einsum`, which fuses the loop over quadrature points $q$ and basis indices $i, j$ into optimized CUDA kernels (e.g., batched GEMM). Crucially, this stage avoids loops over elements, basis functions, and quadrature points, generating a single, compact computation graph node, as illustrated in **Algorithm 1**.

---

**Algorithm 1** Stage 1: Tensorized Batch-Map

---

**Require:** Geometric coordinates $\mathcal{X} \in \mathbb{R}^{E \times k \times d}$, Quadrature rule $(\hat{\mathcal{W}}, \hat{\mathcal{X}}) \in \mathbb{R}^{Q} \times \mathbb{R}^{Q \times d}$, Reference basis $\hat{\mathcal{B}} \in C^{\infty}(\hat{e})^k$.

**Ensure:** Local stiffness tensor $\mathcal{K}_{\text{local}} \in \mathbb{R}^{E \times k \times k}$.
   ▷ Similarly for $\mathcal{F}_{\text{local}} \in \mathbb{R}^{E \times k}$
1: **Geometry:** Compute Jacobians $\mathcal{J}(\mathcal{X})$ and determinants $|\det \mathcal{J}|$ in batch.
2: **Push-forward:** Map gradients $\mathcal{G} \leftarrow \mathcal{J}^{-\top} \cdot \nabla \hat{\mathcal{B}}$.
3: **Physics:** Evaluate coefficients $\mathcal{C}(\mathcal{X}, \hat{\mathcal{X}})$ at physical quadrature points.
4: **Integration:** $\mathcal{K}_{\text{local}} \leftarrow$ `einsum`$(\ldots, \hat{\mathcal{W}}, \mathcal{J}, \mathcal{G}, \mathcal{G}, \mathcal{C})$.
   ▷ Single Kernel
5: **return** $\mathcal{K}_{\text{local}}$

---

### Stage II: Sparse-Reduce (Topology-Aware Routing)

The Reduce stage aggregates local contributions into the global sparse matrix $K$ and load vector $F$. Instead of imperative scatter-add loops, we propose a declarative approach based on *Sparse Matrix Multiplication* (SpMM).

Leveraging assembly linearity, we precompute sparse binary *Routing Matrices* based solely on mesh topology: $\mathcal{S}_{\text{vec}} \in \{0,1\}^{N \times Ek}$ maps flattened element load vectors to $N$ global DoFs, and $\mathcal{S}_{\text{mat}} \in \{0,1\}^{N_{\text{nnz}} \times Ek^2}$ maps stiffness matrices to $N_{\text{nnz}}$ global nonzero entries. Global assembly is thus executed as deterministic sparse projections:

$$F = \mathcal{S}_{\text{vec}} \cdot \text{vec}(\mathcal{F}_{\text{local}}), \quad K = \text{CSR}(\mathcal{I}, \mathcal{S}_{\text{mat}} \cdot \text{vec}(\mathcal{K}_{\text{local}})), \tag{8}$$

where $\mathcal{I}$ denotes precomputed sparse indices. This reformulation replaces millions of atomic scatter-add operations with optimized SpMM kernels, shown in **Algorithm 2**.

---

**Algorithm 2** Stage 2: Unified Sparse-Reduce via SpMM

---

**Require:** Local tensors $\mathcal{K}_{\text{local}} \in \mathbb{R}^{E \times k \times k}, \mathcal{F}_{\text{local}} \in \mathbb{R}^{E \times k}$.
    Routing matrices $\mathcal{S}_{\text{mat}}, \mathcal{S}_{\text{vec}}$.
**Ensure:** Global sparse operator $K$, Global load vector $F$.
  1: **1. Stiffness Matrix Assembly:**
  2:     $m_K \leftarrow \text{vec}(\mathcal{K}_{\text{local}})$     ▷ Flatten local matrices
  3:     $\mathbf{v}_K \leftarrow \mathcal{S}_{\text{mat}} \cdot m_K$     ▷ Aggregate non-zeros
  4:     $K \leftarrow \text{CSR\_Matrix}(\text{indices}, \mathbf{v}_K)$
  5: **2. Load Vector Assembly:**
  6:     $m_F \leftarrow \text{vec}(\mathcal{F}_{\text{local}})$     ▷ Flatten local vectors
  7:     $F \leftarrow \mathcal{S}_{\text{vec}} \cdot m_F$     ▷ Aggregate global nodes
  8: **return** $K, F$

---

**Analysis of the Computational Graph.** The proposed TENSORGALERKIN architecture optimizes the backward pass through two key structural advantages:

- Monolithic Graph Topology: By strictly tensorizing basis interactions, we consolidate the assembly into two monolithic graph nodes ($O(1)$ complexity). This eliminates the $O(k^2)$ fragmentation typical of standard FEM implementations, minimizing Python interpreter and AD engine overhead.

- Zero-Compilation Agility: Leveraging PyTorch's eager execution, TENSORGALERKIN handles dynamic mesh topologies (e.g., in adaptive refinement) without the prohibitive JIT recompilation latency characteristic of XLA-based frameworks such as JAX-FEM.

**Downstream Applications of TENSORGALERKIN.** We visually summarize the TENSORGALERKIN framework in Figure 1. Moreover, we can readily deploy TENSORGALERKIN for the following *downstream tasks*.

*i) Numerical PDE Solver:* Once the Galerkin matrix $K$ has been assembled, standard iterative linear solvers can be used to solve the Matrix Equation (3) classically. We denote the resulting PDE solver as TENSORMESH.

*ii) Physics-informed Operator Learning:* The solution operator $\mathsf{S} : \rho \mapsto u$ of (1) can be learned by minimizing the residual (4) using gradient descent algorithms once the Galerkin matrix $K(\rho)$ has been efficiently computed with TENSORGALERKIN. In contrast to AD-reliant frameworks such as PINNs, VPINNs, and the Deep Ritz method, which rely on `torch.autograd.grad(u, x)` to compute spatial gradients, necessitating multiple backpropagation steps and incurring heavy graph overhead, TENSORGALERKIN computes the spatial derivatives analytically via tensor contraction with pre-calculated shape function gradients $\mathcal{G}$ (see **Algorithm 1** and a concrete example in **SM A.2**). This allows the residual $\mathcal{L}$ in (4) to be evaluated efficiently as a sequence of dense tensor operations and sparse projections, completely bypassing the AD engine for spatial gradient computations and eliminating "graph-within-graph" bottlenecks. We term the resulting framework TENSORPILS, with PILS standing for Physics-Informed Learning System.

*iii) PDE-Constrained Optimization:* Given the variational PDE (1), a typical PDE-constrained optimization or inverse design problem seeks to find

$$\rho^* = \arg\min_{\rho \in \Pi} \Gamma(u, \rho) \text{ s.t. } \mathsf{a}_\rho(u, v) = \ell_\rho(v), \tag{9}$$

for all $v \in W$. Here, $\Gamma$ is an objective function that the design process seeks to minimize in the space of inputs $\rho$, depending on the state $u$, which is constrained to solve the PDE (1) in variational form. These PDE-constrained optimization problems arise in a wide variety of shape and topology optimization, as well as control problems (Hinze et al., 2009).

In practice, the *discrete* version of the optimization problem

$$\rho^* = \arg\min_{\rho \in \Pi} \Gamma(U, \rho) \text{ s.t. } K(\rho)U = F(\rho), \tag{10}$$

is solved to find an approximation of the solution of Eq. (9). Here, $U$ is the discretized Galerkin solution of the underlying variational PDE. We seek to solve the minimization problem (10) using gradient descent methods. These, in turn, require efficient computations of the gradient with respect to the assembled linear system in (10) (Hinze et al., 2009). In classical PDE-constrained design routines, these gradients are computed using the adjoint variable $\lambda$

$$K^\top \lambda = -\frac{\partial \Gamma}{\partial U} \implies \frac{\partial \Gamma}{\partial K} = \lambda U^\top, \frac{\partial \Gamma}{\partial F} = -\lambda. \tag{11}$$

In contrast to classical numerical PDE solvers, where additional adjoint equations need to be formulated and independently solved, leading to significant computational overhead,

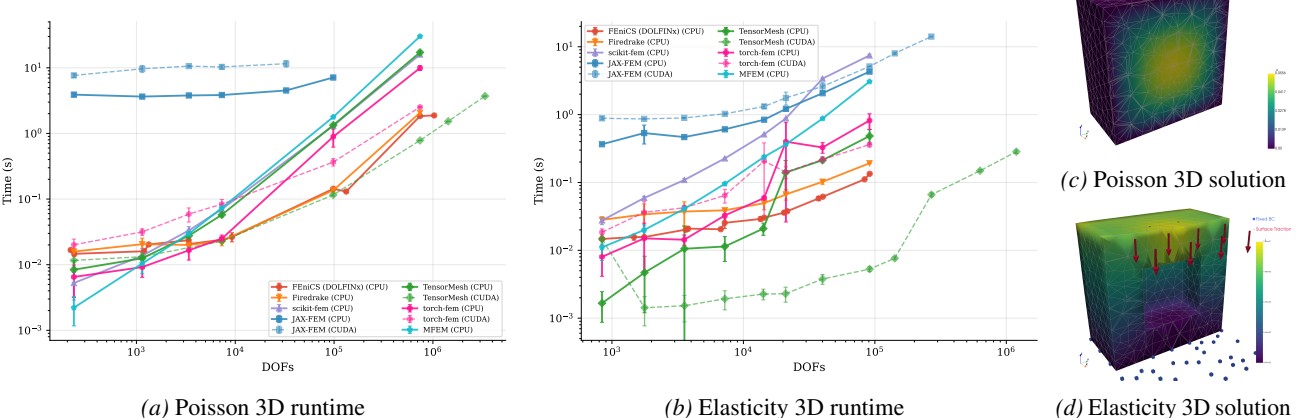

*(a)* Poisson 3D runtime        *(b)* Elasticity 3D runtime        *(d)* Elasticity 3D solution

*Figure 2.* Numerical PDE solver benchmarks on 3D meshes. Panels (a) and (b) report solve-time scaling with the number of DoFs against the FEniCS/Firedrake/MFEM/torch-fem/SKFEM/JAX-FEM baselines. Panels (c) and (d) show representative TENSORMESH solution fields for the Poisson and linear-elasticity problems, respectively; the corresponding residuals against the FEniCS reference are reported in **SM B.1.3**.

TENSORGALERKIN enables this entire pipeline to remain within the PyTorch ecosystem, given the end-to-end differentiable nature of the pipeline. Within it, the adjoint system of Eq. (11) is solved using the same sparse backend, and the gradient products (e.g., $\lambda U^\top$) are computed efficiently without densifying the sparse matrices. Crucially, both the forward operator $K(\rho)$ and its adjoint are assembled through the same TENSORGALERKIN Map–Reduce pipeline, ensuring that gradients with respect to $\rho$ propagate efficiently through the variational discretization without introducing additional adjoint-specific code paths. Concretely, the adjoint solve in Eq. (11) is invoked as a PyTorch-native custom backward pass through the differentiable sparse solver TORCH-SLA (Chi & Wen, 2026); this avoids backpropagating through the BiCGSTAB iterations – which would otherwise instantiate an $O(\text{iter} \times \text{DoFs})$ graph – and keeps the optimization-loop graph at $O(1)$ nodes per iteration. Thus, the entire inverse problem is solved end-to-end within the same TENSORGALERKIN PyTorch framework, providing significant added-value for our method that we term TENSOROPT.

## 3. Results

In this section, we showcase the ability of the TENSOR-GALERKIN framework to very efficiently and accurately handle the following downstream tasks,

**Numerical PDE solver.** To demonstrate the capability of our TENSORMESH as a numerical PDE solver, we consider two PDE benchmarks: the Poisson equation and the linear elasticity equations; see **SM B.1.1** for details on the experimental setup. In Figure 2, we present the runtimes for TENSORMESH as the number of degrees of freedom increases, for both the CPU and GPU versions. As base-

lines, we present runtimes for the extremely popular FEniCS (DOLFINx) framework on CPUs (Baratta et al., 2023), the scikit-fem (SKFEM) framework (Gustafsson & McBain, 2020), Firedrake (Rathgeber et al., 2016), MFEM (Anderson et al., 2021), and the PyTorch-native torch-fem library on CPU and GPU (Meyer, 2024). Runtimes with the XLA based JAX-FEM framework (Xue et al., 2023), on both CPUs and GPUs, are also presented. FEniCS, Firedrake, and MFEM are run with MPI parallelism at $\text{rank} = 8$; full configurations are documented in **SM B.1.1**. We observe from this figure that the GPU version of TENSORMESH is the most efficient algorithm in this context, with the shortest runtimes, even with large numbers ($10^6$) of DoFs, achieving 3 times speedup over the second best-performing baseline (FEniCS) for the Poisson problem and more than one order of magnitude speedup over the second best-performing baseline (TENSORMESH CPU) for the elasticity equations. It is also essential to note that these impressive speedups do not come at the cost of a loss of accuracy. As shown in **SM B.1.3**, the residuals with TENSORMESH are as small (or even smaller than) those of the best-performing baselines. This accuracy is also reflected in the quality of solutions computed with TENSORMESH , as depicted visually in **SM B.1.3**. Furthermore in **SM B.1.4**, we highlight another key attribute of TENSORMESH, i.e., its ability to process large-scale batch data generation, which is essential for many-query problems with PDEs. Beyond pure Dirichlet boundary conditions, TENSORMESH natively handles mixed boundary conditions by routing Neumann and Robin contributions through the same Sparse-Reduce stage; on a Poisson benchmark with mixed Dirichlet+Neumann+Robin BCs (Mousavi et al., 2026), TENSORMESH matches FEniCS to relative error $< 10^{-4}$ while being $52\times$ faster on a circular domain and $18\times$ faster on a non-convex boomerang domain (**SM B.1.5**).

**Neural PDE Solver.** We consider a two-dimensional Poisson equation with *checkerboard* forcing terms at different scales (see **SM B.2.1** for details) to investigate the performance of the proposed TENSORPILS framework as a stand-alone neural PDE solver. We compare its performance with three neural PDE solvers: PINNs (Raissi et al., 2019), VPINNs (Kharazmi et al., 2019), and the Deep Ritz method (E & Yu, 2018), and present the errors in Table 1. These test errors result from a physics-informed (data-free) training of all the models with the following schedule: 10,000 iterations of ADAM, followed by 200 iterations of L-BFGS. Further details of the models are provided in **SM B.2.2**. From Table 1, we observe that TENSORPILS is by far the most accurate on all the forcings that we apply, achieving 50% less error than the nearest baseline (Deep Ritz), while also being 2× faster. These results demonstrate how our use of analytical shape function gradients significantly outperforms the autograd based baselines while still retaining accuracy. From Table 1, we also observe that the PINN framework, which has to perform two AD passes to compute the Laplace operator, is the slowest of all the models while not being more accurate.

*Table 1.* Comparison of neural PDE solvers. We report relative $L^2$ error (%) for different frequencies $K$ and training throughput (it/s). All models were trained for 10,000 Adam + 200 LBFGS steps.

| Method | Rel. $L^2$ Error (%) | | | Speed (it/s) | |
|---|---|---|---|---|---|
| | $K = 2$ | $K = 4$ | $K = 8$ | Adam | LBFGS |
| PINN | 0.90 | 6.30 | 34.77 | 20.1 | 1.0 |
| VPINN | 11.58 | 36.88 | 154.10 | 54.9 | 50.5 |
| Deep Ritz | 3.34 | 4.50 | 10.60 | 58.7 | 55.7 |
| **TENSORPILS** | **0.56** | **2.24** | **10.05** | **117.8** | **99.7** |

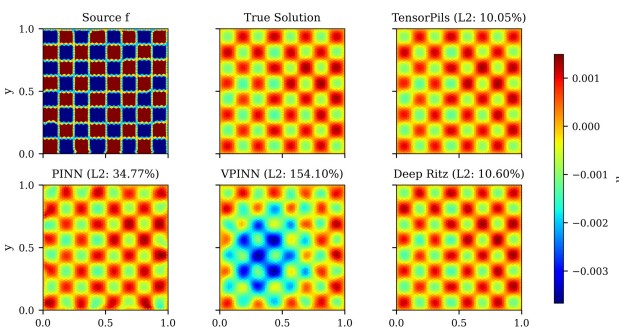

*Figure 3.* Reconstructed solution fields for the $K = 8$ checkerboard forcing. The lower-frequency cases ($K = 2, 4$) are reported in **SM B.2.3**.

Beyond accuracy, we further quantify the scalability of TENSORPILS-style loss evaluation by benchmarking the wall-clock time of a single forward loss computation as a function of the number of DoFs. We compare supervised MSE loss, finite-difference (FD) losses, and PINN losses on regular grids; all methods share the same SIREN backbone and mesh, evaluated on an NVIDIA H200 GPU. As shown in Figure 4, PINN loss evaluation becomes rapidly more expensive as the number of DoFs increases due to AD graph overhead, eventually becoming orders of magnitude slower. In contrast, TENSORPILS exhibits nearly flat scaling and remains close to the cost of finite-difference losses, while retaining applicability to unstructured meshes where stencil-based methods are not available. The same trend persists on unstructured triangular meshes: TENSORPILS is roughly 4–6× faster than the conventional PINN on the forward pass across a $10^3$–$10^7$ DoF sweep, the backward gap reaches $\sim 6\times$ at 2.3M DoFs, and the PINN runs out of memory at 5.8M DoFs while TENSORPILS continues to scale linearly to 11.6M on the same GPU; full curves and benchmark protocol are reported in **SM B.2.4**.

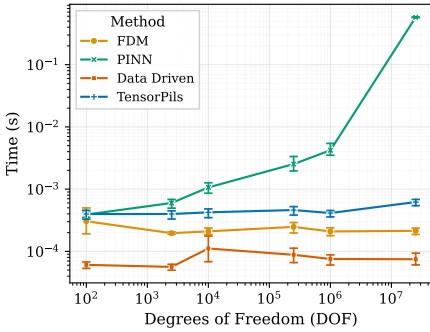

*Figure 4.* CUDA runtime of one forward loss computation vs. DoF for different training objectives on *regular grids*.

**Physics-informed Operator Learning.** In the next set of experiments, we test the entire TENSORPILS pipeline for physics-informed operator learning. To this end, we consider two time-dependent PDEs: the linear acoustic wave equation on a circular domain and the non-linear Allen-Cahn equations on an L-Shaped domain; see further details on the problem setup in **SM B.3.1**. In contrast to the elliptic problems we have considered so far, these PDEs are hyperbolic (wave equation) and parabolic (Allen-Cahn). Moreover, we need to extend the methods we have described in two directions: to time-dependent problems and to include nonlinearities in the underlying PDEs. This entails augmenting the variational form (1) with non-trivial extensions, described in **SM B.3.1**. For both PDEs, our operator learning task amounts to learning the underlying *solution operator* that maps initial conditions to solutions at later times.

Our aim is to learn this solution operator using the TENSORPILS algorithm. To this end, we choose a specific *graph neural network* (GNN) as the parameterized function class $U_\theta$ that maps the input functions into the coefficients of the Galerkin ansatz space (2). GNNs are natural in this context due to the underlying unstructured mesh that discretizes the non-Cartesian geometries. Details of the specific GNN

that we use are provided in **SM B.3.2**. To baseline TEN-SORPILS, we choose two alternatives: i) a fully data-driven method where the ansatz functions (2), with the same GNN backbone as TENSORPILS, are trained in a fully data-driven (and physics-free) manner with 16 training samples, and ii) the physics-informed DeepONet (PI-DeepONet) model of (Wang & Perdikaris, 2021), trained in a data-free manner. The details of the baseline models are provided in **SM B.3.2**. To properly test the model, we consider two test settings:

*Table 2.* Quantitative comparison of physics-informed operator learning performance. We report the relative $L^2$ errors on the wave and Allen-Cahn (AC) equations. The evaluation is performed on both In-distribution (ID) and Out-of-distribution (OOD) test sets.

|  | Wave | | AC | |
|---|---|---|---|---|
|  | ID | OOD | ID | OOD |
| Data-Driven | 0.089±0.013 | 0.230±0.017 | 0.135±0.042 | 0.152±0.080 |
| PI-DeepONet | 0.626±0.033 | 0.863±0.018 | 0.743±0.163 | 8.536±6.306 |
| **TENSORPILS** | **0.085±0.010** | **0.090±0.006** | **0.110±0.014** | **0.083±0.013** |

*In-Distribution* (ID) and *Out-of-distribution* (OOD), with details on both setups provided in **SM B.3.3**. The resulting test errors for the three competing models on both tasks are presented in Table 2. We observe that TENSORPILS is at least an order of magnitude more accurate than PI-DeepONet, which completely fails to approximate the solution operator (see **SM B.3.4** for visualizations). Surprisingly, TENSOR-PILS is even more accurate than the fully data-driven model. This gap in performance is clearly seen in the OOD setting. The data-driven model performs relatively poorly at this extrapolation task with 3× larger errors than in the ID case. On the other hand, the data-free, physics-informed TENSOR-PILS model learns the underlying physics and generalizes to unseen samples at inference, demonstrating the strength of a physics-informed approach for extrapolation in operator learning tasks. We further analyze the effect of varying the number of training samples on both the data-driven and TENSORPILS models in **SM B.3.5**, demonstrating the data efficiency and stability of Galerkin-driven training.

**PDE Constrained Inverse Design.** As a final numerical experiment, we consider a challenging Inverse design from the field of *Topology Optimization*. With the details described in **SM B.4**, our problem can be summarized as a *compliance minimization problem* for a 2D cantilever beam, described by the elasticity equations. Starting from a rectangular design domain, with fixed support on one edge and a point load applied in another corner, we formulate the topology optimization problem in terms of a Solid Isotropic Material with Penalization (SIMP) method (see **SM B.4.1**), reducing the problem to the general form (10) subject to a volumetric constraint and the PDE constraint. We utilize the gradient descent type Method of Moving Asymptotes (MMA) (Svanberg, 1987a) to solve this problem.

*Table 3.* Performance comparison on the 2D Cantilever Beam Topology Optimization task (51 iterations). Time is measured in seconds.

| Stage | JAX-FEM | TENSORMESH(Ours) | Speedup |
|---|---|---|---|
| Setup Time | 2.62 s | **0.58 s** | **4.5×** |
| Optimization Loop | 28.51 s | **7.77 s** | **3.7×** |
| **Total Time** | 31.13 s | **8.35 s** | **3.7×** |

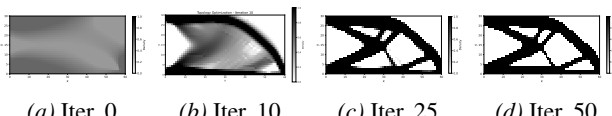

| (a) Iter. 0 | (b) Iter. 10 | (c) Iter. 25 | (d) Iter. 50 |

*Figure 5.* Topology evolution of the 2D cantilever beam during TENSOROPT compliance minimization, from the uniform density initialization (a) to the converged design (d). Details are referred in **SM B.4.2**.

Our baseline utilizes JAX-FEM with the LU decomposition direct solver (internally provided by UMFPACK (Davis, 2004)). In contrast, our proposed model leverages TEN-SOROPT equipped with a BiCGSTAB iterative solver. Crucially, within our framework, sensitivity (the gradient of the objective function with respect to density) is computed via a differentiable solver layer that is fully integrated into the PyTorch autograd graph. The differentiable sparse solver layer is implemented using the TORCH-SLA library (Chi & Wen, 2026), which achieves $O(1)$ extra graph nodes per iteration rather than the $O(\text{iterations} \times \text{DoFs})$ graph nodes that would be incurred by back-propagating through the BiCGSTAB iterations.

We verify that both frameworks converge to topologically identical designs with negligible discrepancies in the final objective function (compliance difference $< 0.33\%$); see **SM B.4.2**. However, as shown in Table 3, TENSORMESH significantly outperforms JAX-FEM in computational efficiency. The setup phase (mesh/problem initialization and compiling setup) is accelerated by $4.5\times$, and the optimization loop, which involves repeated JiT free finite element assembly and solving, achieves a $3.7\times$ speedup. This is because TENSORMESH not only eliminates the Python loop overhead but also provides a highly efficient differentiable substrate for iterative PDE design optimization tasks.

## 4. Discussion

**Summary.** We present a unified algorithmic framework for (numerically) solving, learning, and optimizing PDEs with a variational structure. Based on the observation that the key bottleneck in the execution of current Galerkin-based frameworks on GPUs lies in the assembly of the underlying stiffness matrices and load vectors, we have presented TENSORGALERKIN as an efficient assembly algorithm. TENSORGALERKIN uses a two stage Map-Reduce frame-

work and is engineered for optimal performance within the PyTorch ecosystem. We validate this design by deploying TENSORGALERKIN downstream as i) a numerical PDE solver TENSORMESH, ii) a physics-informed operator learning framework TENSORPILS, and iii) an end-to-end differentiable PDE-constrained optimization framework TENSOROPT. We demonstrate the performance of these frameworks in terms of accuracy and computational efficiency on a series of PDE benchmarks, observing that our proposed frameworks are significantly more efficient than state-of-the-art baselines while retaining or increasing accuracy. Thus, we provide a flexible and high-performance algorithmic foundation for solving, learning, and optimizing PDE-based systems.

**Related Work.** Recalling that TENSORGALERKIN provides an integrated algorithmic framework for PDE learning, numerical solutions, and optimization, we should emphasize that, to the best of the authors' knowledge, there is no alternative single algorithmic framework with the same level of generality and flexibility. However, there are multiple frameworks that cover one or more of the downstream tasks that we target with TENSORGALERKIN.

In terms of the numerical solution of PDEs, the backbone method of TENSORGALERKIN is a well-known Galerkin-type FEM discretization. However, our novelty lies in the efficient algorithmic realization of the assembly process. The two assembly primitives we rely on — batched `einsum` contractions and SpMM-based sparse aggregation — are themselves standard, the latter with a long history in algebraic multigrid and graph-BLAS workloads (Ruge & Stüben, 1987; Buluç & Gilbert, 2011); what is new is their co-design inside PyTorch's autograd, yielding the $O(1)$-graph property. In this regard, TENSORMESH can be compared to finite element software frameworks such as FEniCS (Baratta et al., 2023), Firedrake (Rathgeber et al., 2016), MFEM (Anderson et al., 2021), SKFEM (Gustafsson & McBain, 2020), and torch-fem (Meyer, 2024), which are widely used. In contrast to these frameworks, TENSORMESH is fully GPU compatible, and we also show that it is an order of magnitude faster to run than them (Figure 2). The JAX-FEM platform (Xue et al., 2023) is a notable example of a modern GPU compatible finite element framework. We have compared TENSORMESH extensively to JAX-FEM and demonstrate that TENSORMESH is orders of magnitude faster on the same hardware.

As a neural PDE solver, TENSORPILS can be compared to methods such as PINNs (Raissi et al., 2019), VPINNs (Kharazmi et al., 2019), weak PINNs (De Ryck et al., 2022), and Deep Ritz (E & Yu, 2018), as well as their numerous variants; see a summary in (De Ryck & Mishra, 2024). All these methods rely on automatic differentiation (through (multiple) backpropagation passes) to compute spa-

tial (and temporal) gradients, leading to large computational overheads. On the other hand, we completely decouple the spatial gradient information from the neural networks in TENSORPILS and use analytical shape gradients to efficiently compute these gradients, thereby removing these overheads. This allows TENSORPILS to be much more computationally efficient while significantly improving the accuracy of PINN type methods.

In terms of physics-informed operator learning, TENSORPILS can be compared to existing frameworks such as PI-DeepONet (Wang & Perdikaris, 2021) and PINO (Li et al., 2024). In contrast to these frameworks which rely on automatic differentiation for spatial derivative computations and lead to computational overheads, TENSORPILS utilizes the TENSORGALERKIN machinery for efficient derivative computation, significantly increasing efficiency. Moreover, PINO is restricted to Cartesian grids. While PI-DeepONet can handle general (but fixed) input geometries, it performs poorly on the experiments that we present here, being less accurate by at least an order of magnitude while taking significantly longer to train than TENSORPILS. In terms of the underlying method, TENSORPILS overlaps with the methods in (Gao et al., 2022; Berrone et al., 2022) but expands them to the operator learning setting while also providing a very different and highly efficient implementation.

Finally, TENSOROPT utilizes the end-to-end differentiable pipeline of TENSORGALERKIN to provide a differentiable PDE solver for inverse design problems such as topology optimization. In this space, while differentiable PDE solvers such as JAX-FEM exist, we show that TENSOROPT is significantly more efficient while being equally accurate.

**Limitations and Future Work.** Our main structural assumption is that the underlying PDE possesses a variational structure (1). While a large number of PDEs are variational, many do not possess the same structure, which limits the applicability of TENSORGALERKIN. However, we plan to extend our framework to non-conforming finite-element methods such as DG and Petrov-Galerkin methods to address an extended set of PDEs in the future. Similarly, further testing on three-dimensional PDE systems in complex domain geometries is planned for the future, as is an extensive investigation of time-stepping techniques, which we cover very briefly in this work. Further downstream applications to PDE-constrained control and optimization problems in more real-world settings are also envisaged.

## Impact Statement

This paper presents work whose goal is to advance the field of machine learning, particularly at the intersection of differentiable numerical methods, physics-informed learning, and scientific computing. There are many potential societal consequences of such techniques, none of which we feel must be specifically highlighted here.

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

## A. Problem Formulation and Variational Structures

### A.1. General Time-Dependent and Semi-Linear Framework

While the main text focuses on the abstract linear operator equation $\mathsf{a}_\rho(u, v) = \ell_\rho(v)$, many physical systems of interest are inherently time-dependent and may involve nonlinearities. Our TENSORGALERKIN framework naturally extends to these settings via the *Method of Lines*.

Consider a time-dependent PDE for $u(t, x)$ on $\Omega \times [0, T]$. The semi-discrete variational formulation seeks $u(t) \in V$ such that for almost every $t \in (0, T]$:

$$\langle \partial_t u, v \rangle_M + \mathsf{a}_\rho(u, v) + \mathcal{N}_\rho(u; v) = \ell_\rho(v), \quad \forall v \in W, \tag{A.1}$$

where:

- $\langle \cdot, \cdot \rangle_M$ represents the mass inner product (typically $L^2(\Omega)$), leading to the mass matrix $M$.

- $\mathsf{a}_\rho(\cdot, \cdot)$ is the generic bilinear form (e.g., stiffness/diffusion), leading to the stiffness matrix $K(\rho)$.

- $\mathcal{N}_\rho(u; v)$ represents semi-linear forms that are non-linear in $u$ but linear in the test function $v$ (e.g., reaction terms like $u^3$).

- $\ell_\rho(v)$ is the linear functional for external sources.

Discretizing the spatial domain with the Galerkin ansatz $u_h(t, x) = \sum_{j=1}^N U_j(t)\phi_j(x)$ leads to a system of Ordinary Differential Equations (ODEs):

$$M\dot{U}(t) + K(\rho)U(t) + F_{\text{nonlin}}(U(t)) = F_{\text{ext}}(t), \tag{A.2}$$

where TENSORGALERKIN is employed to efficiently assemble $M, K(\rho)$, and the nonlinear residual vectors $F_{\text{nonlin}}$ at each required step (or once, if constant).

### A.2. A Concrete PDE Example

We explicitly define the bilinear forms and functionals in (1) for the following linear scalar elliptic equation

$$\begin{aligned} -\nabla \cdot (\rho(\mathbf{x})\nabla u(\mathbf{x})) &= f(\mathbf{x}) && \text{in } \Omega, \\ u(\mathbf{x}) &= 0 && \text{on } \partial\Omega, \end{aligned} \tag{A.3}$$

where $\Omega \in \mathbb{R}^2$ is a bounded polygonal domain, $\rho : \Omega \to \mathbb{R}$ is the diffusion coefficient, and $f : \Omega \to \mathbb{R}$ is the source function. Let $V := H_0^1(\Omega)$ be the Sobolev space containing square integrable functions with square integrable gradients and zero trace on $\partial\Omega$. The variational formulation of Eq. (A.3) seeks $u \in V$ such that

$$\underbrace{\int_\Omega \rho(\mathbf{x})\nabla u(\mathbf{x}) \cdot \nabla v(\mathbf{x})\, \mathrm{d}\mathbf{x}}_{:=\mathsf{a}_\rho(u,v)} = \underbrace{\int_\Omega f(\mathbf{x})v(\mathbf{x})\, \mathrm{d}\mathbf{x}}_{:=\ell(v)} \quad \forall\, v \in V. \tag{A.4}$$

Partition the domain $\Omega$ by a triangular mesh, namely, $\Omega = \cup_{m=1}^E e_m$, where $e_m \subset \Omega$ is the $m$-th triangular element. Denote by $\{\mathbf{z}_i\}_{i=1}^N$ the collection of nodes in the interior of $\Omega$. If using the first-order Lagrangian element, the DoFs are associated with the interior nodes, and the basis function $\{\phi_i\}_{i=1}^N$ is piecewise linear and satisfies

$$\phi_i(\mathbf{z}_j) = \delta_{ij} \quad \forall\, i, j \in \{1, \ldots, N\}. \tag{A.5}$$

Note that the basis function $\phi_i$ is nonzero only on triangles that share the node $\mathbf{z}_i$. Define $V_h := \text{span}\{\phi_i\}_{i=1}^N \subset V$. The finite element discretization of Eq. (A.4) seeks $u_h \in V_h$ such that

$$\mathsf{a}_\rho(u_h, v_h) = \ell(v_h) \quad \forall\, v_h \in V_h. \tag{A.6}$$

Plugging in the ansatz $u_h(\mathbf{x}) = \sum_{j=1}^N U_j \phi_j(\mathbf{x})$ and the definition of $\mathbf{a}_\rho(\cdot, \cdot)$ and $\ell(\cdot)$, it amounts to seeking $U :=$ $[U_1, \ldots, U_N]^\top$ such that

$$\sum_{j=1}^N \int_\Omega \rho(\mathbf{x})\nabla\phi_j(\mathbf{x}) \cdot \nabla\phi_i(\mathbf{x}) \, \mathrm{d}\mathbf{x} \cdot U_j = \int_\Omega f(\mathbf{x})\phi_i(\mathbf{x}) \, \mathrm{d}\mathbf{x} \quad \forall i \in \{1, \ldots, N\}. \tag{A.7}$$

In view of (3), we end up with the following matrix equation

$$KU = F \quad \text{where } K_{ij} = \int_\Omega \rho(\mathbf{x})\nabla\phi_j(\mathbf{x}) \cdot \nabla\phi_i(\mathbf{x}) \, \mathrm{d}\mathbf{x}, \quad F_i = \int_\Omega f(\mathbf{x})\phi_i(\mathbf{x}) \, \mathrm{d}\mathbf{x}. \tag{A.8}$$

In what follows, we elaborate on the proposed Map-Reduce algorithm (see **Algorithms 1** and **2**) for this concrete example.

In **Stage I (Batch-Map)**, we compute the collection of local Galerkin matrices $\mathcal{K}_{\mathrm{local}} := \{K_e\}_{e=1}^E, K_e \in \mathbb{R}^{k \times k}$ (see Eq. (5) for the definition of $K_e$). Recall that $k$ denotes the number of DoFs associated with a single element. In the current setting, $k = 3$, since the local DoFs lie on the three nodes of a triangle. Recall the local-global DoFs mapping $g_e : \{1, \ldots, k\} \to \{1, \ldots, N\}$. Given an element $e$, we denote $\phi_a^{(e)}, a = 1, 2, 3$ as the local basis function on $e$ associated with the $a$-th node of $e$, that is, $\phi_a^{(e)}(\mathbf{x}) = \phi_{g_e(a)}(\mathbf{x})$ for $\mathbf{x} \in e$. From the implementation viewpoint, we treat each element $e$ as the image of a diffeomorphism $\Phi_e : \hat{e} \to e$, with $\hat{e} := \{x \geq 0, y \geq 0, x + y \leq 1\}$ being the reference element, on which we define the reference basis linear functions $\{\hat\phi_a\}_{a=1}^3$ such that $\hat\phi_a(\hat{\mathbf{x}}_b) = \delta_{ab}$, where $\{\hat{\mathbf{x}}_a\}_{a=1}^3$ are the three nodes of $\hat{e}$. The global basis functions $\{\phi_i\}_{i=1}^N$ are then simply characterized through the *push-forward* of the reference basis functions $\{\hat\phi_a\}_{a=1}^3$. Namely, it holds that

$$\phi_{g_e(a)}(\Phi_e(\hat{\mathbf{x}})) = \phi_a^{(e)}(\Phi_e(\hat{\mathbf{x}})) = \hat\phi_a(\hat{\mathbf{x}}) \quad \forall \hat{\mathbf{x}} \in \hat{e}. \tag{A.9}$$

Such a setup offers immense implementation convenience, allowing us to evaluate all the integrals in a uniform manner through a quadrature rule $\{(\hat{w}_q, \hat{\mathbf{x}}_q)\}_{q=1}^Q$ defined on $\hat{e}$. Specifically, each local Galerkin matrix $K_e$ can be computed as

$$(K_e)_{ab} := \int_e \rho(\mathbf{x})\nabla\phi_b^{(e)}(\mathbf{x}) \cdot \nabla\phi_a^{(e)}(\mathbf{x}) \, \mathrm{d}\mathbf{x} = \int_{\hat{e}} \rho(\Phi_e(\hat{\mathbf{x}})) \left(J_e^{-\top}(\hat{\mathbf{x}})\nabla\hat\phi_b(\hat{\mathbf{x}})\right) \cdot \left(J_e^{-\top}(\hat{\mathbf{x}})\nabla\hat\phi_a(\hat{\mathbf{x}})\right) |\det J_e(\hat{\mathbf{x}})| \, \mathrm{d}\hat{\mathbf{x}} \tag{A.10}$$

where $J_e(\hat{\mathbf{x}}) := \mathrm{D}\Phi_e(\hat{\mathbf{x}}) \in \mathbb{R}^{2 \times 2}$ is the Jacobian of $\Phi_e$ at $\hat{\mathbf{x}} \in \hat{e}$. Similarly, the local load vector can be computed as

$$(F_e)_a := \int_e f(\mathbf{x})\phi_a^{(e)}(\mathbf{x}) \, \mathrm{d}\mathbf{x} = \int_{\hat{e}} f(\Phi_e(\hat{\mathbf{x}}))\hat\phi_a(\hat{\mathbf{x}})|\det J_e(\hat{\mathbf{x}})| \, \mathrm{d}\hat{\mathbf{x}}. \tag{A.11}$$

Note that in this example $J_e$ is a constant matrix on $\hat{e}$ due to the affine mapping $\Phi_e$ from the reference triangle to physical triangles, but $J_e$ varies with respect to $\hat{\mathbf{x}}$ in general cases. Applying the quadrature rule yields

$$(K_e)_{ab} \simeq \sum_{q=1}^Q \hat{w}_q \cdot \rho(\Phi_e(\hat{\mathbf{x}}_q)) \left(J_e^{-\top}(\hat{\mathbf{x}}_q)\nabla\hat\phi_b(\hat{\mathbf{x}}_q)\right) \cdot \left(J_e^{-\top}(\hat{\mathbf{x}}_q)\nabla\hat\phi_a(\hat{\mathbf{x}}_q)\right) |\det J_e(\hat{\mathbf{x}}_q)|, \qquad a, b \in \{1, \ldots, k\},$$

$$(F_e)_a \simeq \sum_{q=1}^Q \hat{w}_q \cdot f(\Phi_e(\hat{\mathbf{x}}_q))\hat\phi_a(\hat{\mathbf{x}}_q)|\det J_e(\hat{\mathbf{x}}_q)|, \qquad a \in \{1, \ldots, k\}. \tag{A.12}$$

In the current example, the batch quantities involved in **Algorithm 1** are encoded as follows:

$$\begin{aligned}
\mathcal{X} &\leftarrow \Phi_e(\hat{\mathbf{x}}_a), & e \in \{1, \ldots, E\}, \, a \in \{1, \ldots, k\}, \\
\hat{\mathcal{W}} &\leftarrow \hat{w}_q, & q \in \{1, \ldots, Q\}, \\
\hat{\mathcal{X}} &\leftarrow \hat{\mathbf{x}}_q, & q \in \{1, \ldots, Q\}, \\
\hat{\mathcal{B}} &\leftarrow \hat\phi_a, & a \in \{1, \ldots, k\}, \\
\mathcal{J} &\leftarrow J_e(\hat{\mathbf{x}}_q), & e \in \{1, \ldots, E\}, \, q \in \{1, \ldots, Q\}, \\
\mathcal{G} &\leftarrow J_e^{-\top}(\hat{\mathbf{x}}_q)\nabla\hat\phi_a(\hat{\mathbf{x}}_q), & e \in \{1, \ldots, E\}, \, q \in \{1, \ldots, Q\}, \, a \in \{1, \ldots, k\}, \\
\mathcal{C} &\leftarrow \rho(\Phi_e(\hat{\mathbf{x}}_q)), & e \in \{1, \ldots, E\}, \, q \in \{1, \ldots, Q\}, \\
\mathcal{K}_{\mathrm{local}} &\leftarrow K_e, & e \in \{1, \ldots, E\}, \\
\mathcal{F}_{\mathrm{local}} &\leftarrow F_e, & e \in \{1, \ldots, E\}.
\end{aligned} \tag{A.13}$$

In view of Eq. (A.12), the function $\mathcal{F}$ in Eq. (7) takes the following explicit expression:

$$\mathcal{F}\left(\mathcal{G}_{eqa}, \mathcal{G}_{eqb}, \mathcal{C}_{eq}\right) := \rho(\Phi_e(\hat{\mathbf{x}}_q)) \left( J_e^{-\top}(\hat{\mathbf{x}}_q)\nabla\hat{\phi}_b(\hat{\mathbf{x}}_q)\right) \cdot \left( J_e^{-\top}(\hat{\mathbf{x}}_q)\nabla\hat{\phi}_a(\hat{\mathbf{x}}_q)\right). \tag{A.14}$$

As emphasized in the algorithm, all the computations are fully-tensorized, that is, suitable contractions are performed without any loops. Noted that the local vectors $\mathcal{F}_{\text{local}}$ are computed via an analogous, yet computationally simpler, rank-1 tensor contraction.

In **Stage II (Sparse-Reduce)**, entries in $\mathcal{K}_{\text{local}}$ and $\mathcal{F}_{\text{local}}$ are aggregated into the global Galerkin matrix $K$ and load vector $F$. We realize the scatter-add operation defined in Eq. (6) via sparse projections using the precomputed routing matrices $\mathcal{S}_{\text{mat}}$ and $\mathcal{S}_{\text{vec}}$. This step does not depend on physics (that is, the underlying PDE to be solved) and has been illustrated clearly in **Algorithm 2**.

# B. Experiments

## B.1. Numerical PDE Solver

### B.1.1. PROBLEM SETUP

We evaluate the efficiency and numerical fidelity of TENSORMESH on two three-dimensional elliptic benchmarks and compare against standard FEM stacks (FEniCS (Baratta et al., 2023), scikit-fem (Gustafsson & McBain, 2020), and JAX-FEM (Xue et al., 2023)). Unless otherwise stated, all methods use tetrahedral meshes and piecewise linear ($\mathbb{P}_1$) basis functions.

**Benchmark I: 3D Poisson Equation.** On the unit cube $\Omega = [0, 1]^3$, we solve

$$\begin{cases} -\Delta u(\mathbf{x}) = f(\mathbf{x}) & \text{in } \Omega, \\ u(\mathbf{x}) = 0 & \text{on } \partial\Omega, \end{cases} \tag{B.1}$$

with constant source $f = 1$ and homogeneous Dirichlet boundary conditions on all faces.

**Benchmark II: 3D Linear Elasticity.** We consider the static equilibrium of an isotropic linear elastic body:

$$\begin{cases} -\nabla \cdot \boldsymbol{\sigma}(\mathbf{x}) = \mathbf{f}(\mathbf{x}) & \text{in } \Omega, \\ \mathbf{u}(\mathbf{x}) = \mathbf{0} & \text{on } \partial\Omega, \end{cases} \tag{B.2}$$

where $\mathbf{u} : \Omega \to \mathbb{R}^3$ is the displacement and the Cauchy stress satisfies Hooke's law

$$\boldsymbol{\sigma} = \lambda \operatorname{tr}(\boldsymbol{\varepsilon})\, \mathbf{I} + 2\mu\,\boldsymbol{\varepsilon}, \quad \boldsymbol{\varepsilon} = \tfrac{1}{2}(\nabla\mathbf{u} + (\nabla\mathbf{u})^\top). \tag{B.3}$$

We set Young's modulus $E = 1$ and Poisson's ratio $\nu = 0.3$, yielding

$$\lambda = \frac{E\nu}{(1+\nu)(1-2\nu)}, \quad \mu = \frac{E}{2(1+\nu)}. \tag{B.4}$$

A constant body force $\mathbf{f} = (1, 1, 1)^\top$ is applied. To introduce geometric complexity, we use a hollow cube domain

$$\Omega = [0, 1]^3 \setminus (0.25, 0.75)^3. \tag{B.5}$$

### B.1.2. BASELINE SOLVER CONFIGURATION

To ensure a controlled comparison across all FEM frameworks (including TENSORMESH), we use a unified linear solver configuration: BiCGSTAB (van der Vorst, 1992) with Jacobi (diagonal) preconditioning. Table B.1 summarizes the common solver parameters.

For GPU-accelerated solvers on problems exceeding 200,000 DoF, we employ cuDSS (NVIDIA Corporation, 2024) when available and fall back to the iterative method otherwise. The convergence criterion is

$$\frac{\|\mathbf{K}\mathbf{u} - \mathbf{f}\|}{\|\mathbf{f}\|} < \epsilon_{\text{rel}}, \quad \epsilon_{\text{rel}} = 10^{-10}, \tag{B.6}$$

where $\mathbf{K}$ and $\mathbf{f}$ denote the condensed stiffness matrix and the load vector after applying Dirichlet boundary conditions.

*Table B.1.* Unified linear solver configuration for all FEM baseline frameworks.

| Parameter | Value |
|---|---|
| Iterative method | BiCGSTAB |
| Preconditioner | Jacobi (diagonal scaling) |
| Relative tolerance | $10^{-10}$ |
| Absolute tolerance | $10^{-10}$ |
| Maximum iterations | 10,000 |

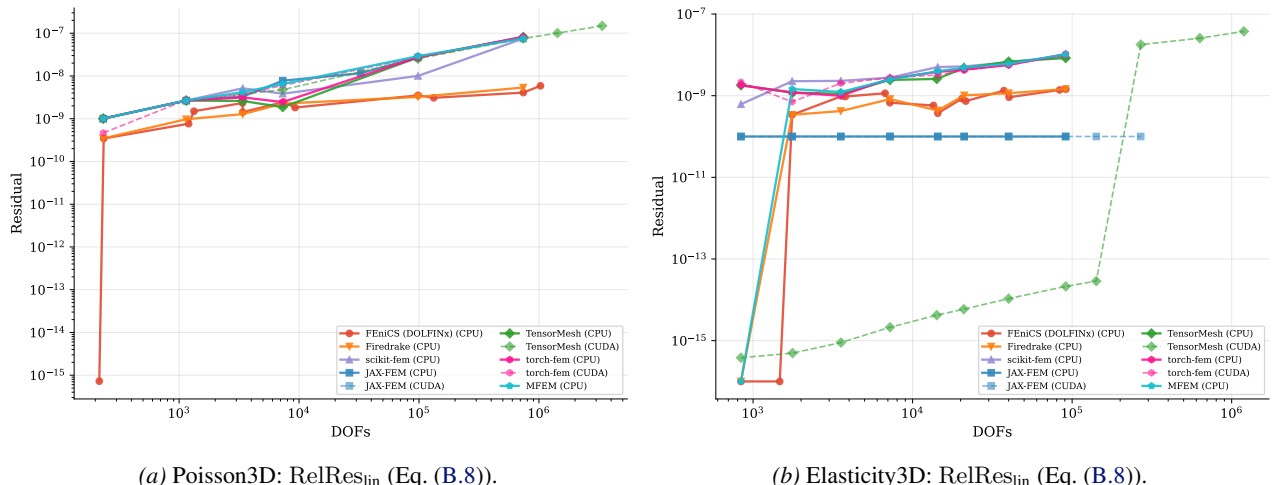

*(a)* Poisson3D: $\text{RelRes}_{\text{lin}}$ (Eq. (B.8)).  *(b)* Elasticity3D: $\text{RelRes}_{\text{lin}}$ (Eq. (B.8)).

*Figure B.1.* Relative linear-system residual vs. degrees of freedom (DoF) for 3D Poisson and 3D elasticity.

**Physics-Informed Neural Network (PINN) Setup**    We additionally include PINNs as a learning-based baseline that minimizes the *strong-form* PDE residual via gradient-based training. We use a SIREN MLP (Sitzmann et al., 2020) (3 hidden layers, width 64, $\omega = 30$) with 1 output (Poisson) or 3 outputs (elasticity). The objective is $\mathcal{L} = \mathcal{L}_{\text{PDE}} + \lambda_{\text{BC}} \mathcal{L}_{\text{BC}}$ with $\lambda_{\text{BC}} = 100$, where $\mathcal{L}_{\text{PDE}}$ is the mean squared residual on interior collocation points and $\mathcal{L}_{\text{BC}}$ enforces Dirichlet boundary conditions on boundary samples. Training uses Adam for 1,000 epochs (cosine schedule $10^{-3} \rightarrow 10^{-5}$, gradient clipping 1.0) followed by 50 L-BFGS iterations (strong Wolfe). For each mesh resolution, the PINN is trained from scratch and all spatial derivatives are computed via automatic differentiation.

### B.1.3. ERROR ANALYSIS AND VISUALIZATION

We report (i) relative solution error, (ii) residual of the discretized linear system, and (iii) qualitative visualizations.

**Reference solution and metrics.**    Let $u_{\text{ref}}$ denote the TENSORMESH solution under the same solver tolerance, chosen as a numerical reference because it achieves the smallest residual among the compared methods. We measure the relative error by

$$\text{RelErr}(u) := \frac{\|u - u_{\text{ref}}\|_2}{\|u_{\text{ref}}\|_2}. \tag{B.7}$$

For both Poisson3D and Elasticity3D, we report the relative linear-system residual

$$\text{RelRes}_{\text{lin}}(u) := \frac{\|Ku - f\|_2}{\|f\|_2}, \tag{B.8}$$

where $K$ and $f$ denote the condensed stiffness matrix and load vector after applying Dirichlet boundary conditions.

**Residual scaling with degrees of freedom.**    Figure B.1 summarizes residuals versus DoF for both Poisson3D and Elasticity3D. Across both problems, TENSORMESH matches (and often improves upon) the residual level of established FEM toolchains while supporting GPU execution via TENSORGALERKIN. In contrast, PINN does not exhibit comparable residual decay under refinement in these 3D settings.

**PINN error/residual table.** Table B.2 reports PINN's error and residual under increasing mesh resolution. For Poisson3D, PINN reduces the relative error with refinement but remains far from satisfying the linear-system residual at FEM-level accuracy. For Elasticity3D, PINN fails to reduce the linear-system residual and shows limited error improvement, consistent with Figure B.1.

*Table B.2.* PINN error and residual on 3D Poisson and 3D Elasticity problems under mesh refinement.

| Problem | chara. length | DoF(s) | RelErr | $\text{RelRes}_{\text{lin}}$ |
|---|---|---|---|---|
| Poisson3D | 0.20 | 233 | 1.1027 | 1.0998 |
| Poisson3D | 0.10 | 1145 | 0.0292 | 0.2423 |
| Poisson3D | 0.07 | 3392 | 0.0155 | 0.2027 |
| Poisson3D | 0.05 | 7315 | 0.0115 | 0.2083 |
| Elasticity3D | 0.20 | 840 | 0.9922 | 1.0003 |
| Elasticity3D | 0.10 | 3549 | 0.9163 | 1.0000 |
| Elasticity3D | 0.07 | 10563 | 0.8180 | 0.9999 |

**Qualitative visualization.** Figure B.2 and Figure B.3 visualize representative solutions produced by different solvers. TENSORMESH is visually consistent with established FEM baselines, whereas PINN exhibits noticeable artifacts and reduced physical fidelity, aligning with its larger residuals.

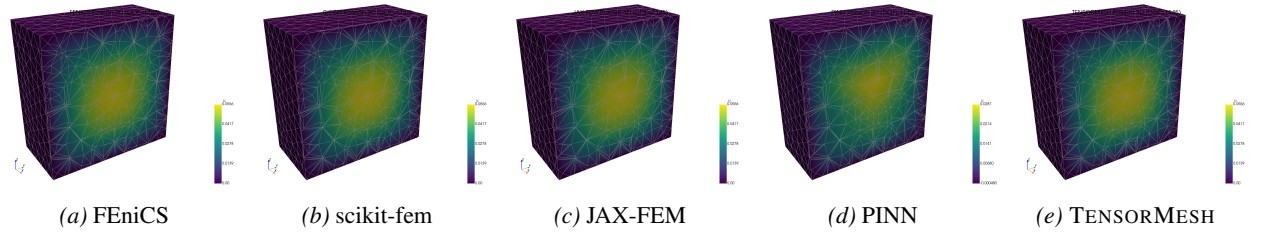

|          |             |            |          |               |
|:--------:|:-----------:|:----------:|:--------:|:-------------:|
| *(a)* FEniCS | *(b)* scikit-fem | *(c)* JAX-FEM | *(d)* PINN | *(e)* TENSORMESH |

*Figure B.2.* Poisson3D solution visualization across solvers.

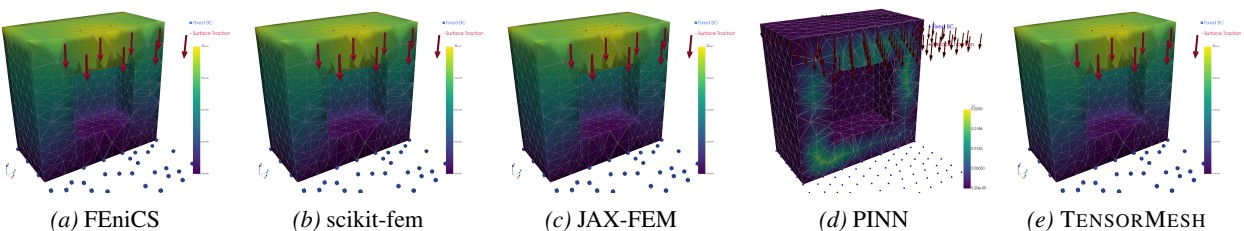

|          |             |            |          |               |
|:--------:|:-----------:|:----------:|:--------:|:-------------:|
| *(a)* FEniCS | *(b)* scikit-fem | *(c)* JAX-FEM | *(d)* PINN | *(e)* TENSORMESH |

*Figure B.3.* Elasticity3D solution visualization across solvers.

### B.1.4. EFFICIENT BATCH DATA GENERATION

A distinct advantage of TENSORMESH is its ability to accelerate batched data generation for SciML workflows. Training pipelines frequently require solving the same PDE operator on a fixed mesh with varying source terms or boundary conditions (e.g., to create large $(f, u)$ datasets). We evaluate this regime using a 3D Poisson equation on a mesh with 7,315 degrees of freedom (DoFs), keeping the mesh topology fixed while varying the right-hand side $f$. Figure B.4 illustrates the generation time as a function of batch size. While the CPU baseline exhibits super-linear scaling (slope $\approx 1.15$), the CUDA-accelerated TENSORMESH demonstrates distinct overhead amortization: for batch sizes up to $10^2$, the runtime remains nearly constant, indicating that the fixed overhead dominates computation. Even at large scales, the runtime grows with a slope of $0.92$, significantly outperforming the CPU. This throughput efficiency makes TENSORMESH ideal for constructing massive physics-based datasets.

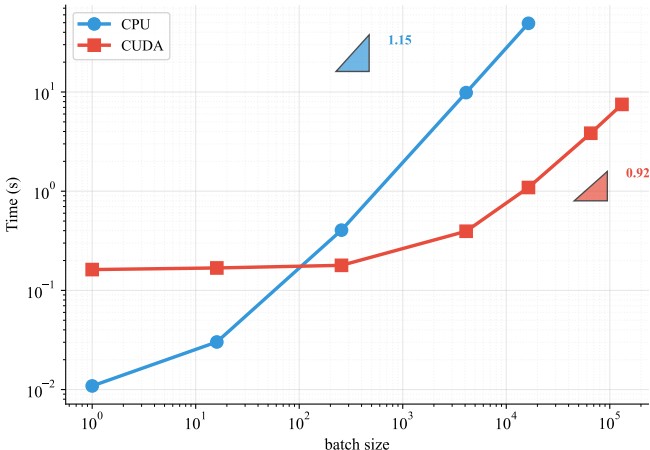

*Figure B.4.* Efficient batch data generation.

### B.1.5. MIXED BOUNDARY CONDITIONS ON NON-CONVEX GEOMETRIES

We solve a Poisson benchmark from (Mousavi et al., 2026) with simultaneously enforced *Dirichlet*, *Neumann*, and *Robin* boundary conditions on a circular and a non-convex *boomerang* domain (analytical solution available). Within TENSORGALERKIN, the Neumann and Robin boundary integrals are routed through the same Map–Reduce pipeline used for volumetric integrals (a batched `einsum` over boundary quadrature followed by a sparse boundary-routing projection); no special-case code paths are introduced. Both frameworks use $\mathbb{P}_1$ Lagrange elements on the same mesh. End-to-end (assembly + solve) CPU wall-clock times are reported in Table B.3; TENSORMESH solutions are shown in Figure B.5.

*Table B.3.* Mixed Dirichlet + Neumann + Robin Poisson benchmark. End-to-end (assembly + solve) time on a single CPU.

| Dataset | Mesh | FEniCSx (CPU) | TENSORMESH (CPU) | Speedup |
|---------|------|---------------|------------------|---------|
| Poisson circle (bc5) | 6K nodes | 7,000 ms | 133 ms | ∼**52×** |
| Poisson boomerang (bc5) | 14.8K nodes | 5,600 ms | 317 ms | ∼**18×** |

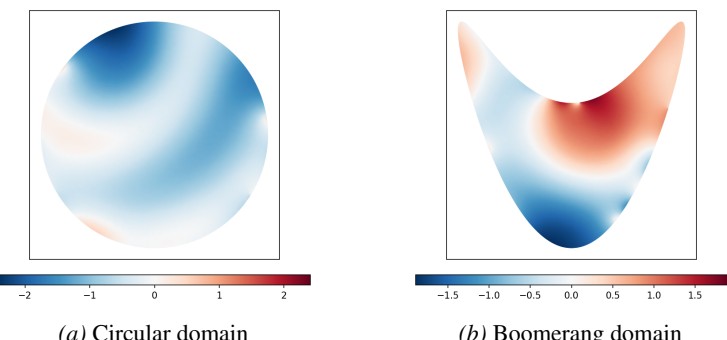

*(a)* Circular domain      *(b)* Boomerang domain

*Figure B.5.* TENSORMESH solution to the mixed Dirichlet+Neumann+Robin Poisson benchmark on (a) a circular domain and (b) a non-convex boomerang domain.

## B.2. Neural PDE Solver

In this section, we evaluate TENSORPILS as a standalone *neural PDE solver* in a single-instance learning setting. We compare against three physics-informed baselines: PINN (strong-form residual), VPINN (variational residual), and Deep Ritz (energy minimization). To isolate the effect of the learning paradigm, all methods share the same SIREN backbone and optimizer hyperparameters, and differ only in their objective $\mathcal{L}(\theta)$ and boundary-condition handling.

### B.2.1. PROBLEM SETUP

We consider the 2D Poisson equation on $\Omega = [0,1]^2$ with homogeneous Dirichlet boundary conditions:

$$\begin{cases} -\Delta u(x,y) = f_K(x,y), & (x,y) \in \Omega, \\ u(x,y) = 0, & (x,y) \in \partial\Omega. \end{cases} \tag{B.9}$$

To stress-test high-frequency behavior and discontinuities, we use a checkerboard forcing

$$f_K(x,y) = (-1)^{\lfloor Kx \rfloor + \lfloor Ky \rfloor}, \tag{B.10}$$

which alternates between $\pm 1$ on a $K \times K$ grid and introduces discontinuities along $x = i/K$ and $y = j/K$. As $K$ increases, the solution exhibits increasingly multi-scale structures, making the task challenging under spectral bias. Figure B.6 visualizes $f_K$ for $K \in \{1, 2, 4, 8\}$. Ground-truth solutions for error reporting are computed using a high-fidelity FEM solver on a fine mesh.

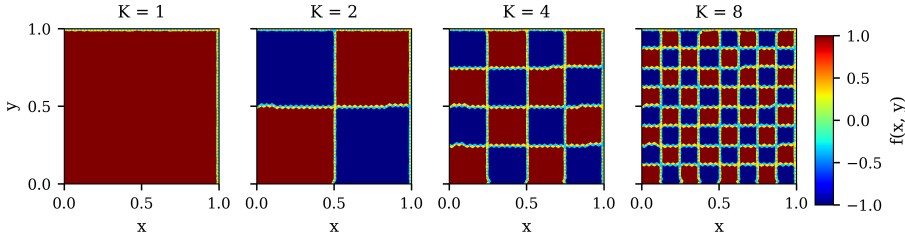

*Figure B.6.* Checkerboard forcing $f_K(x,y)$ for $K \in \{1, 2, 4, 8\}$.

### B.2.2. MODEL AND LEARNING SETUP

We enforce a strict control setting: all methods use the same network backbone, mesh, and optimizer schedule; only the physics-informed objective and constraint implementations differ.

**Backbone.** We use a SIREN (Sitzmann et al., 2020) with input $\mathbf{x} \in \mathbb{R}^2$, 4 hidden layers of width 64, and sine activations with $\omega_0 = 30$:

$$\mathbf{z}_0 = \mathbf{x}, \tag{B.11}$$

$$\mathbf{z}_i = \sin\big(\omega_0(\mathbf{W}_i \mathbf{z}_{i-1} + \mathbf{b}_i)\big), \quad i = 1, \dots, 4, \tag{B.12}$$

$$u_\theta(\mathbf{x}) = \mathbf{W}_{\text{out}} \mathbf{z}_4 + \mathbf{b}_{\text{out}}. \tag{B.13}$$

We follow the SIREN initialization in (Sitzmann et al., 2020) to ensure stable optimization.

**Shared discretization.** All experiments use the same unstructured triangular mesh with 3,017 nodes and 6,036 elements. Each method evaluates its loss on the same geometric information (mesh nodes and/or element quadrature points), enabling a controlled comparison.

**Learning paradigms.** Figure B.7 summarizes the four objectives:

- PINN (strong form). Minimize pointwise residual $\|\Delta u_\theta + f_K\|^2$ on mesh nodes plus a soft boundary penalty $\lambda_{\text{bc}} \|u_\theta\|^2_{\partial\Omega}$; requires second-order AD.

- VPINN (variational form). Minimize a variational residual using test functions and numerical quadrature; uses first-order AD for $\nabla u_\theta$; boundary conditions via soft penalty.

- Deep Ritz (energy). Minimize the energy functional $J(u_\theta)$ with deterministic Gaussian quadrature on elements (rather than Monte Carlo); boundary conditions via soft penalty.

- TENSORPILS (ours). Predict FEM coefficients $U$ and minimize the discrete residual $\|\mathbf{K}\mathbf{U} - \mathbf{F}\|^2$. Dirichlet boundary conditions are imposed as hard constraints by reducing the linear system, and all spatial derivatives are computed analytically from shape functions (no AD).

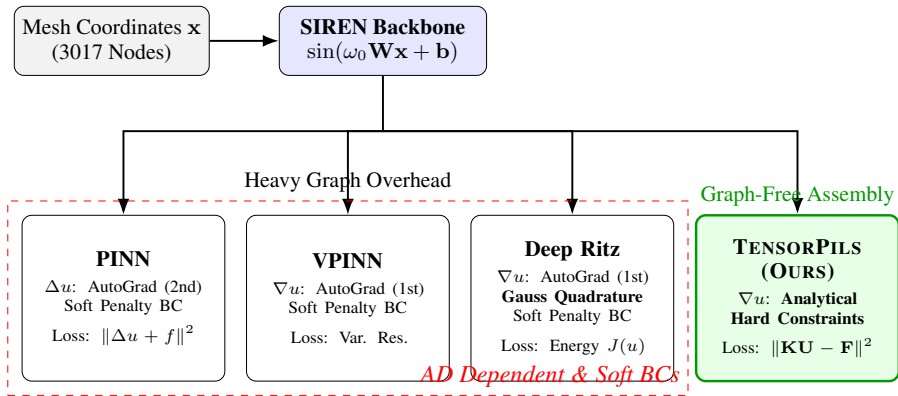

*Figure B.7.* Schematic comparison of learning paradigms under a controlled setting (shared SIREN backbone and mesh).

### B.2.3. VISUALIZATION

We visualize the learned solution fields for increasing forcing frequency $K \in \{2, 4, 8\}$. Figures B.8–B.10 show the forcing, ground truth, and predictions from all four methods, highlighting the robustness of TENSORPILS on multi-scale/discontinuous forcing.

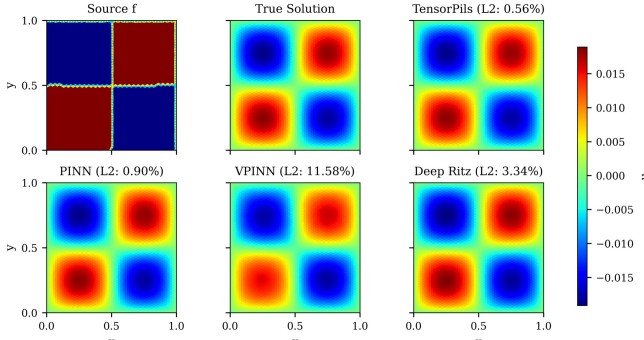

*Figure B.8.* Low-frequency case ($K = 2$).

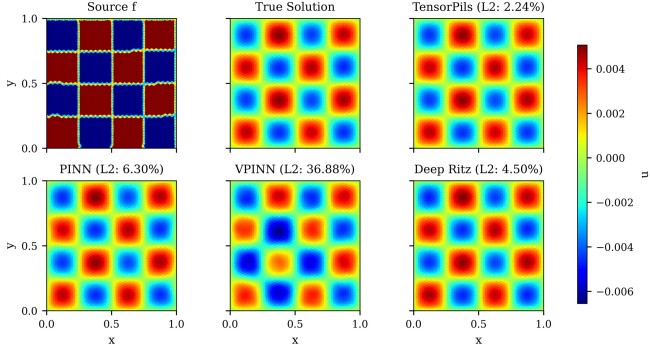

*Figure B.9.* Medium-frequency case ($K = 4$).

Figure B.11 reports the training curves for the most challenging case ($K = 8$). All methods are trained with 10,000 Adam epochs followed by 200 L-BFGS epochs. Since the objectives differ (strong residual vs. variational residual vs. energy vs. discrete residual), loss magnitudes are not directly comparable; we therefore also report the MSE to the ground truth during training.

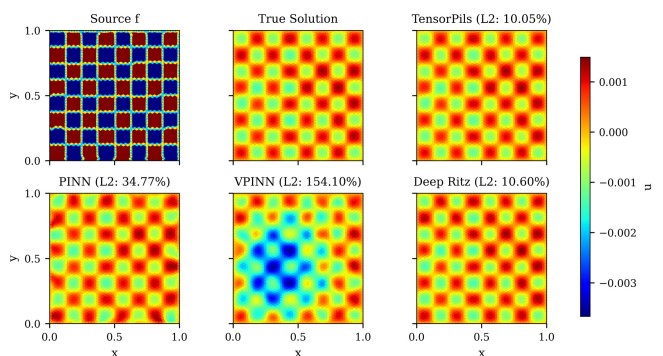

*Figure B.10.* High-frequency case ($K = 8$).

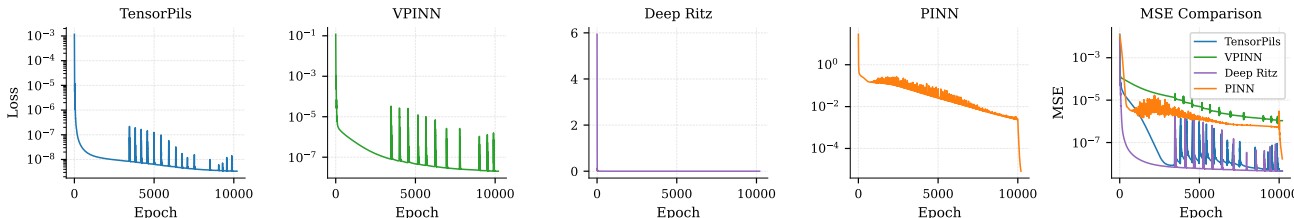

*Figure B.11.* Training curves for $K = 8$: method-specific loss and MSE to the FEM ground truth.

### B.2.4. LOSS-EVALUATION COST ON UNSTRUCTURED MESHES

The regular-grid benchmark in Figure 4 of the main paper establishes that TENSORPILS tracks the finite-difference baseline on Cartesian grids while PINN becomes orders of magnitude slower. We now extend the benchmark to *unstructured triangular meshes*, where stencil-based finite-difference losses are no longer applicable, and additionally report the *backward* pass — the regime where AD-based training is most expensive. We compare a supervised (data-driven) baseline, TENSORPILS, and the PINN loss; all methods share the same SIREN backbone and a $\mathbb{P}_1$ FEM discretization, and we sweep the number of DoFs across four orders of magnitude on a single NVIDIA H200 GPU.

Figure B.12 reports the resulting forward and backward timings. On the *forward* pass, TENSORPILS is roughly 4–6× faster than the PINN loss throughout the entire $10^3$–$10^7$ DoF range and tracks the data-driven baseline closely, while all three losses scale linearly with the number of DoFs. The *backward* pass amplifies the gap: at 2.3M DoFs PINN already takes 73.65 ms versus 12.27 ms for TENSORPILS ($\sim 6\times$), and the PINN autograd graph runs out of memory at 5.8M DoFs, whereas TENSORPILS and the data-driven baseline continue to scale linearly up to 11.6M on the same GPU. This behaviour is a direct consequence of the $O(1)$-graph property of TENSORGALERKIN: PINN instantiates one autograd node per quadrature point per spatial derivative and dominates peak memory as the mesh refines, whereas TENSORGALERKIN routes every assembly through two SpMM-shaped graph nodes regardless of mesh size.

### B.3. Physics-informed Operator Learning

#### B.3.1. PROBLEM SETUP

We study physics-informed *operator learning* for time-dependent PDEs on unstructured meshes. The learning goal is to approximate a solution operator that maps randomized initial conditions to the full spatio-temporal trajectory, and to compare TENSORPILS against a purely data-driven baseline and a physics-informed neural operator baseline (PI-DeepONet).

**Hyperbolic PDE: Wave Equation.** We consider the 2D wave equation on a spatial domain $\mathcal{D} \subset \mathbb{R}^2$ and time interval $[0, T]$:

$$\partial_{tt} u = c^2 \Delta u, \quad \text{in } \mathcal{D} \times [0, T], \tag{B.14}$$

with homogeneous Dirichlet boundary conditions. The initial condition is a multi-frequency sine expansion:

$$u_0(x, y) = \frac{\pi}{K^2} \sum_{i,j} a_{ij} (i^2 + j^2)^{-r} \sin(\pi i x) \sin(\pi j y), \tag{B.15}$$

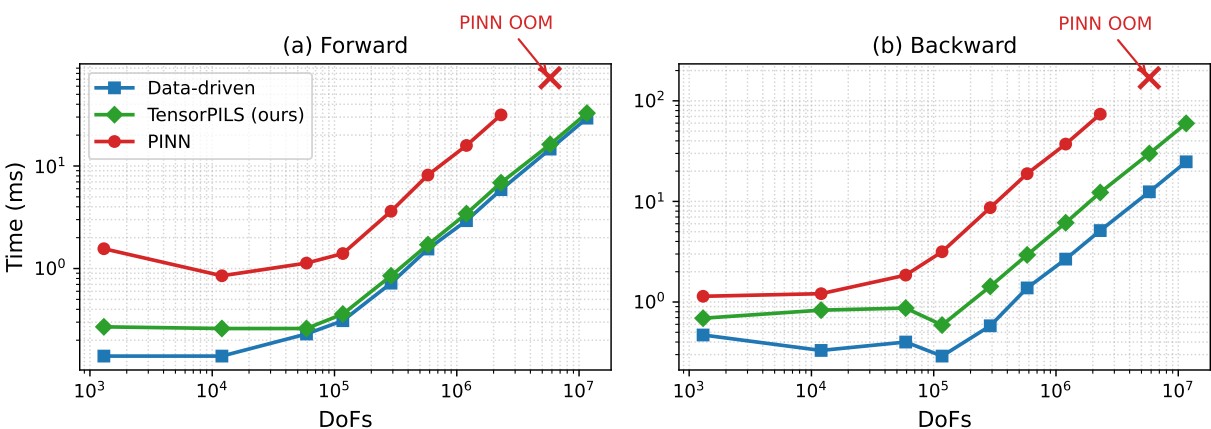

*Figure B.12.* Forward (a) and backward (b) wall-clock time of a single loss evaluation on *unstructured triangular meshes* for the data-driven, TENSORPILS (ours), and PINN losses, as the number of DoFs grows from $10^3$ to $\sim 10^7$. PINN runs out of memory at 5.8M DoFs while TENSORPILS and the data-driven baseline scale linearly to 11.6M on the same H200 GPU.

where $K = 6$, $r = 0.5$, $c = 4$, and $a \in \mathbb{R}^{K \times K}$ is sampled i.i.d. from $\mathcal{U}[-1, 1]$. We use a circular domain centered at $(0.5, 0.5)$ with radius $0.5$.

**Variational discretization and residual.**    Let $\{\phi_j\}$ be $\mathbb{P}_1$ FEM basis functions and $U^k$ be the coefficient vector at time step $t_k$. Using standard Galerkin discretization in space and a second-order central difference in time, the wave dynamics can be written in a matrix form:

$$\mathbf{M}\frac{\mathbf{U}^{k+2} - 2\mathbf{U}^{k+1} + \mathbf{U}^k}{\Delta t^2} + c^2 \mathbf{K}\mathbf{U}^{k+1} = \mathbf{0}, \tag{B.16}$$

where $\mathbf{M}$ and $\mathbf{K}$ denote the FEM mass and stiffness matrices (with Dirichlet rows/cols condensed), as discussed in **SM A**. We define the per-step discrete residual

$$\mathbf{R}^k_{\text{wave}} := \mathbf{M}\frac{\mathbf{U}^{k+2} - 2\mathbf{U}^{k+1} + \mathbf{U}^k}{\Delta t^2} + c^2 \mathbf{K}\mathbf{U}^{k+1}. \tag{B.17}$$

We generate reference trajectories using a stable FEM time integrator satisfying CFL constraints; we use a Crank–Nicolson-style scheme to improve stability and energy behavior.

**Parabolic PDE: Allen–Cahn Equation.**    We further consider the Allen–Cahn (AC) equation:

$$\partial_t u = \nabla \cdot (a^2 \nabla u) - \varepsilon^2 u(u^2 - 1), \quad \text{in } \mathcal{D} \times (0, T), \tag{B.18}$$

with homogeneous Dirichlet boundary conditions. The initial condition is constructed as in Eq. (B.15). The hyper-parameters controlling initial conditions are sampled i.i.d. from $\mathcal{U}[-1, 1]$ with $K = 6$ and $r = 0.5$.

**Variational discretization and residual.**    Using backward Euler in time and Galerkin discretization in space, we obtain a residual form

$$\mathbf{R}^k_{\text{ac}} := \frac{1}{\Delta t}\mathbf{M}(\mathbf{U}^{k+1} - \mathbf{U}^k) + a^2 \mathbf{K}\mathbf{U}^{k+1} - \mathbf{F}(\mathbf{U}^{k+1}), \tag{B.19}$$

where $\mathbf{F}(\mathbf{U})$ denotes the FEM load vector induced by the reaction term $-\varepsilon^2 u(u^2 - 1)$. For the FEM ground truth, we use backward Euler for temporal discretization for the reference solver.

B.3.2. MODEL SETUP

We compare three operator learning paradigms under the same training/test trajectories: (i) data-driven supervised learning, (ii) PI-DeepONet (physics-informed neural operator baseline), and (iii) TENSORPILS (ours).

**AGN (Autoregressive Graph Network).** We follow an autoregressive graph network (AGN) backbone for rollout prediction on unstructured meshes. The mesh is converted into an element graph by connecting nodes within each element as a fully connected subgraph (Figure B.13). The AGN follows the encoder–processor–decoder design commonly used in mesh-based simulators (Pfaff et al., 2020; Brandstetter et al., 2022). We use GraphSAGE (Hamilton et al., 2017) as the processor and employ frequency-enhanced MLPs for the encoder/decoder:

$$X'_v = \left[ X_v, \sin\left(\tfrac{1}{K}X_v\right), \cos\left(\tfrac{1}{K}X_v\right), \dots, \sin(KX_v), \cos(KX_v) \right]. \tag{B.20}$$

Where $X_v$ denotes the initial input for each vertex. Hyperparameter $K$ determines the range and granularity of the spectral features. With window size $w$, the AGN predicts bundled updates and performs time integration to produce multi-step rollouts. Dirichlet constraints are enforced by clamping boundary nodes after each step.

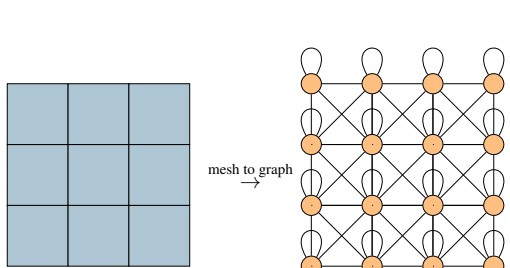

*Figure B.13.* Element graph: each element is a fully connected subgraph.

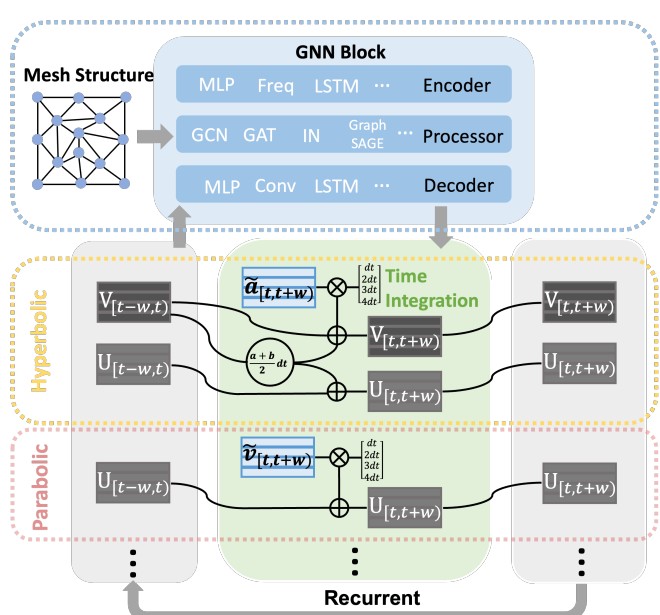

*Figure B.14.* AGN solver with window size $w$ and $n$ rollout steps.

**Data-driven baseline.** The data-driven baseline trains the same AGN using supervised loss on FEM trajectories:

$$\mathcal{L}_{\text{data}} = \sum_{k \in \mathcal{T}_{\text{train}}} \|\widehat{\mathbf{U}}^k - \mathbf{U}^k\|_2^2, \tag{B.21}$$

where $\mathbf{U}^k$ is the FEM ground truth and $\widehat{\mathbf{U}}^k$ is the predicted state.

**TENSORPILS (ours).** TENSORPILS trains the AGN by minimizing the discrete variational residual evaluated by TENSOR-GALERKIN. For wave and AC, we use the per-step residuals in Eq. (B.17) and Eq. (B.19):

$$\mathcal{L}_{\text{gal}} = \sum_{k \in \mathcal{T}_{\text{train}}} \|\mathbf{R}^k\|_2^2, \tag{B.22}$$

where $\mathbf{R}^k$ is the assembled discrete residual vector. All spatial derivatives are computed analytically through shape functions, avoiding AD overhead.

**PI-DeepONet (baseline).** We adopt PI-DeepONet as an operator-learning baseline, consistent with our discussion in the main text. PI-DeepONet consists of a branch net and a trunk net: the *branch* takes the sampled initial condition values at sensor points (we use all mesh nodes) and produces a latent code, and the *trunk* takes spatio-temporal coordinates $(x, y, t)$ (covering all training time steps and all training physical points) to produce a coordinate-dependent embedding. The output is their inner product (plus bias) predicting $u(x, y, t)$. Both branch/trunk are MLPs with activation `tanh`, $n_{\text{hidden}} = 64$, and $n_{\text{layers}} = 4$.

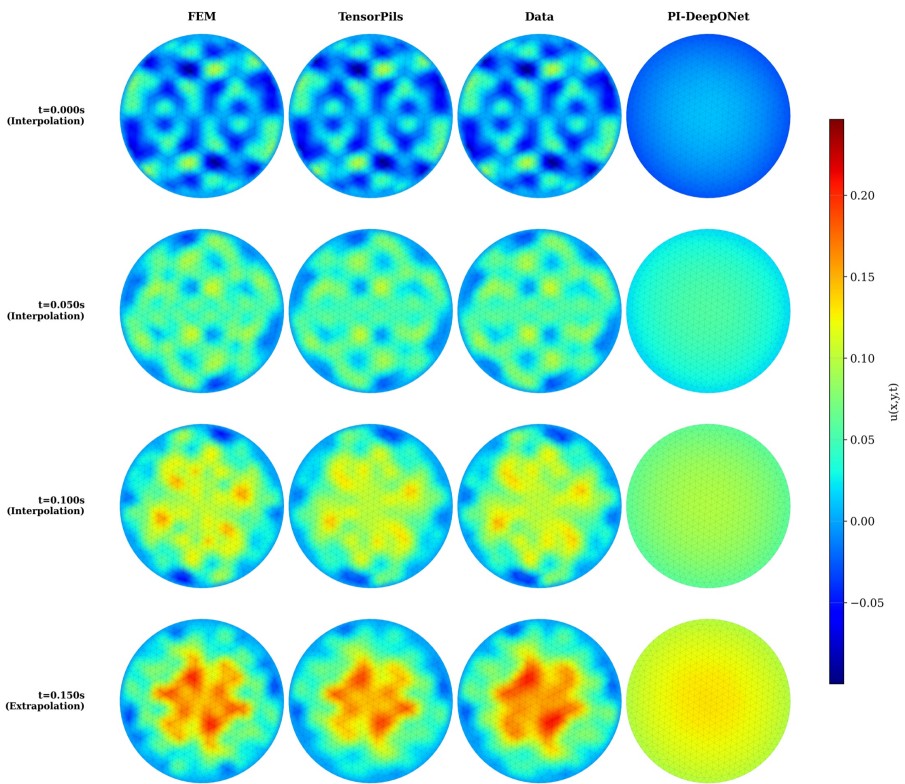

*Figure B.15.* Wave equation: FEM ground truth vs. TENSORPILS, data-driven, and PI-DeepONet at $t = \{0.000, 0.025, 0.050, 0.075\}$s.

PI-DeepONet is trained with a PINN-style physics loss on collocation points:

$$\mathcal{L}_{\text{PI-DeepONet}} = \mathcal{L}_{\text{PDE}} + \lambda_{\text{BC}}\mathcal{L}_{\text{BC}} \tag{B.23}$$

where $\mathcal{L}_{\text{PDE}}$ enforces the strong-form residual of Eq. (B.14) or Eq. (B.18) (computed via AD), and $\mathcal{L}_{\text{BC}}$ enforces Dirichlet boundary constraints.

### B.3.3. EXPERIMENT SETUP

To explore unstructured domains, we use Gmsh to generate triangular meshes. Table B.5 summarizes mesh statistics and time steps.

<table>
<tr><td colspan="2">

*Table B.4.* Experimental Platform Configuration

</td></tr>
</table>

| Component | Specification |
|---|---|
| CPU | Threadripper 3970X (32-Core) |
| GPU | NVIDIA RTX 3090 (24GB) |
| CUDA Version | 11.4 |
| PyTorch | 2.0.1+cu118 |

*Table B.5.* Information for Dataset Setup

| Metric | Wave | Allen–Cahn |
|---|---|---|
| $\|\mathcal{C}\|$ (Elements) | 1185 | 734 |
| $\|\mathcal{V}\|$ (Nodes) | 633 | 408 |
| Shape | Circle | L-Shape |
| $\Delta t$ | $5 \times 10^{-4}$ | $1 \times 10^{-4}$ |

We set $K = 6$ and $r = 0.5$ in Eq. (B.15). For wave dynamics, we additionally include an initial velocity. We set the initial velocity to zero, i.e., $\partial_t u(\cdot, 0) = 0$. For the AGN, we use a window size $w = 4$ and train with a rollout horizon of 200 time steps. We sample 16 random initial conditions to form the training set and evaluate on held-out initial conditions. FEM trajectories are used to (i) train the data-driven baseline and (ii) evaluate errors for all methods.

To properly assess generalization, we report results under two test settings: an In-Distribution (ID) setting and an Out-of-Distribution (OOD) setting. For each initial condition, we simulate trajectories of length 400 time steps and split them into

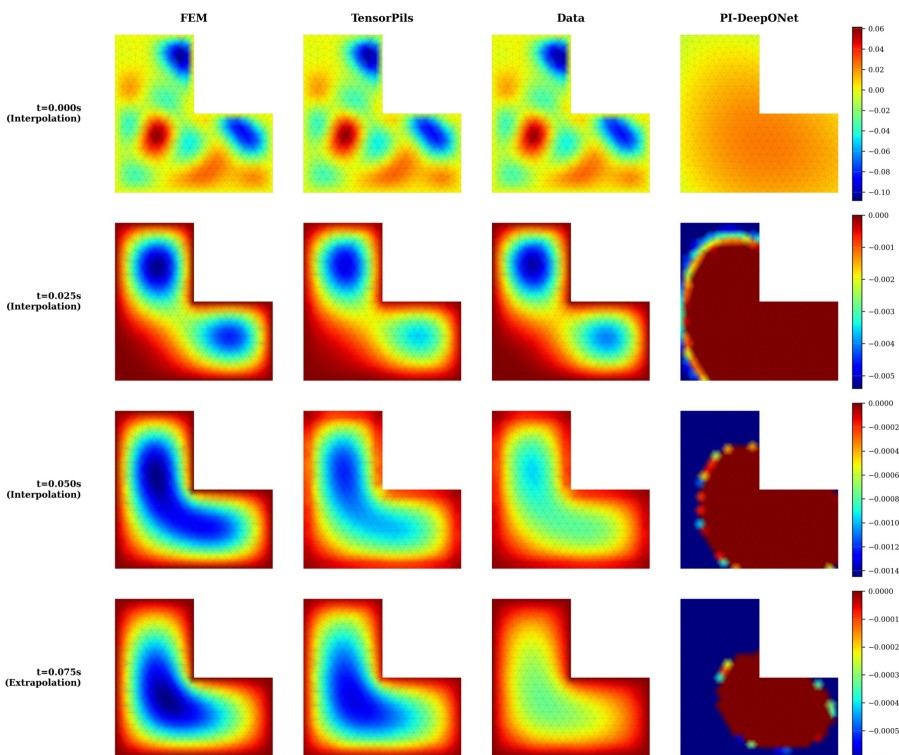

*Figure B.16.* Allen–Cahn equation: FEM ground truth vs. TENSORPILS, data-driven, and PI-DeepONet at $t = \{0.000, 0.050, 0.100, 0.150\}$s.

two consecutive segments. The first 200 steps (t = 1,…,200) are treated as the ID regime, since they match the temporal horizon observed during training. The last 200 steps (t = 201,…,400) are treated as the OOD regime, as they evaluate longer-horizon rollout behavior beyond the training horizon. During training, we only use the first 200 steps from each trajectory (for all initial conditions), and we never expose the model to steps beyond t = 200. At test time, we evaluate performance separately on the ID segment (steps 1–200) and the OOD segment (steps 201–400), reporting the corresponding metrics for each regime.

**Training.** Unless otherwise stated, AGN-based models are trained for 4,000 epochs with Adam (learning rate $10^{-3}$) and a step scheduler (decay factor 0.8 every 500 epochs). The GraphSAGE processor uses hidden size 64 with 3 layers, and $K = 4$ for frequency-enhanced MLPs of encoder and decoder defined in Eq. B.20. PI-DeepONet uses the same learning rate schedule; its PDE residual is computed via AD.

### B.3.4. VISUALIZATION

We provide qualitative comparisons and long-horizon error accumulation for both PDE families. We visualize (i) FEM ground truth, (ii) TENSORPILS, (iii) data-driven, and (iv) PI-DeepONet.

**Wave equation.** Figure B.15 visualizes solutions at $t = \{0.000, 0.025, 0.050, 0.075\}$ seconds.

**Allen–Cahn snapshots.** Figure B.16 visualizes solutions at $t = \{0.000, 0.050, 0.100, 0.150\}$ seconds.

**Wave error accumulation.** To assess rollout stability, Figure B.17 reports the per-step RMSE and accumulated RMSE over time for wave rollouts.

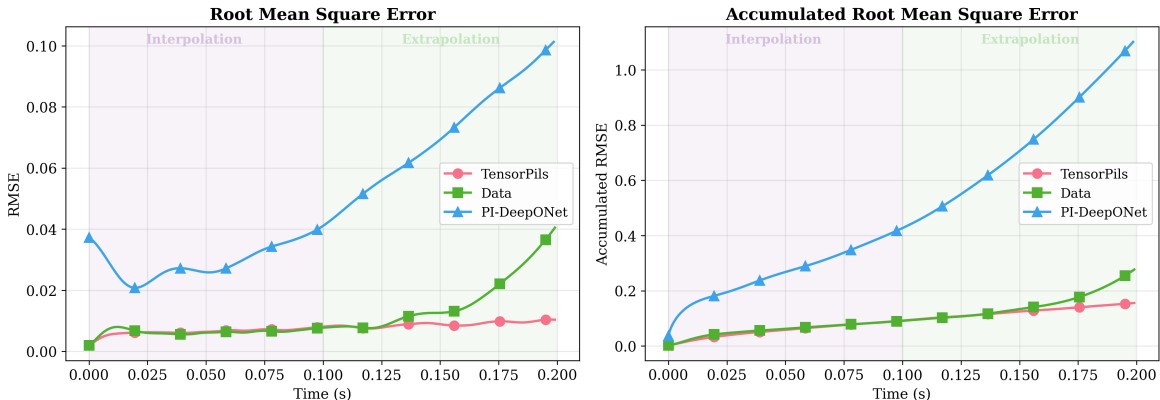

*Figure B.17.* Wave equation error accumulation: (left) per-step RMSE; (right) accumulated RMSE over rollout time.

### B.3.5. FURTHER EXPERIMENTS

We further analyze the data-efficiency and stability of Galerkin-driven training by TENSORPILS on wave dynamics by varying the number of training initial conditions while keeping the AGN architecture fixed. Each model is evaluated on 50 random test initial conditions. As depicted in Figure B.18, the model trained with Galerkin loss exhibits more efficient learning, achieving robustness with fewer training datasets. Remarkably, even with just a single dataset, the AGN trained with Galerkin loss maintains a low error rate of approximately 10%.

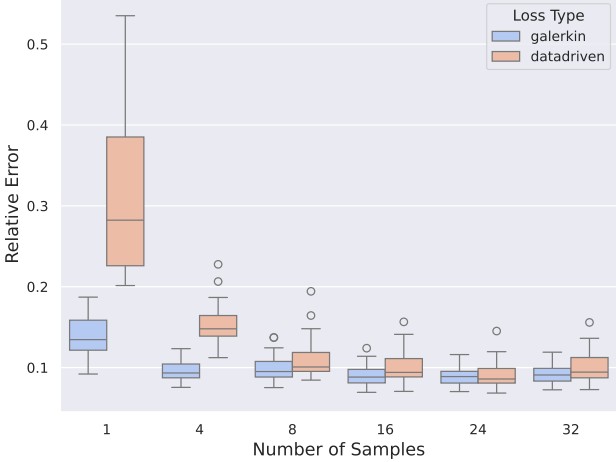

*Figure B.18.* $L^2$ (mean $\pm$ std) relative errors on test dataset for different number of training samples.

## B.4. PDE-Constrained Inverse Design

### B.4.1. PROBLEM SETUP

We demonstrate the differentiability and efficiency of TENSOROPT, built upon the TENSORGALERKIN and TENSORMESH pipeline, through a classical topology optimization benchmark: compliance minimization of a 2D cantilever beam using the Solid Isotropic Material with Penalization (SIMP) method.

**Geometry and Boundary Conditions.** The computational domain is a rectangular region $\Omega = [0, L_x] \times [0, L_y]$ with $L_x = 60$ and $L_y = 30$ (dimensionless units). The domain is discretized using a structured mesh of $60 \times 30$ bilinear quadrilateral elements (QUAD4), resulting in 1,891 nodes and 1,800 elements. Homogeneous Dirichlet boundary conditions

are imposed on the left edge:

$$\mathbf{u} = \mathbf{0} \quad \text{on } \Gamma_D = \{(x, y) : x = 0\}, \tag{B.24}$$

and a distributed traction load is applied to a portion of the right boundary:

$$\mathbf{t} = [0, -100]^\top \text{ N/m} \quad \text{on } \Gamma_N = \{(x, y) : x = L_x, \, 0 \le y \le 0.1L_y\}. \tag{B.25}$$

**SIMP Material Interpolation.** The material stiffness is parameterized using the SIMP interpolation scheme:

$$E(\rho) = E_{\min} + \rho^p(E_{\max} - E_{\min}), \tag{B.26}$$

where $\rho \in [\rho_{\min}, 1]$ is the element-wise density (design variable), $p = 3$ is the penalization exponent promoting binary (0-1) solutions, $E_{\max} = 70{,}000$ MPa is the solid material stiffness, and $E_{\min} = 70$ MPa prevents stiffness matrix singularity. The Poisson's ratio is $\nu = 0.3$.

**Optimization Formulation.** The topology optimization problem seeks to minimize the structural compliance (maximize stiffness) subject to a volume constraint:

$$
\begin{aligned}
\min_{\boldsymbol{\rho}} \quad & C(\boldsymbol{\rho}) = \mathbf{u}^\top \mathbf{K}(\boldsymbol{\rho})\mathbf{u} = \mathbf{F}^\top \mathbf{u} \\
\text{s.t.} \quad & \frac{1}{|\Omega|} \int_\Omega \rho \, d\Omega \le \bar{v} \\
& \mathbf{K}(\boldsymbol{\rho})\mathbf{u} = \mathbf{F} \\
& \rho_{\min} \le \rho_e \le 1, \quad \forall e
\end{aligned}
\tag{B.27}
$$

where $\bar{v} = 0.5$ is the target volume fraction and $\rho_{\min} = 10^{-3}$.

**Sensitivity Analysis.** The gradient of the compliance with respect to the design variables is classically derived using the adjoint method. For SIMP interpolation, the resulting element-wise sensitivity admits the closed-form expression:

$$\frac{\partial C}{\partial \rho_e} = -p\rho_e^{p-1}(E_{\max} - E_{\min}) \, \mathbf{u}_e^\top \mathbf{K}_0^e \mathbf{u}_e, \tag{B.28}$$

where $\mathbf{K}_0^e$ denotes the element stiffness matrix evaluated at unit Young's modulus. In TENSORMESH, however, this gradient is *not explicitly implemented*. Instead, it is obtained automatically via PyTorch's reverse-mode automatic differentiation, by backpropagating through the differentiable assembly and sparse solve. The above expression is reported for reference and to highlight consistency with classical topology optimization formulations.

**Optimization Algorithm.** We employ the Method of Moving Asymptotes (MMA) (Svanberg, 1987b) with a move limit of $\Delta\rho_{\max} = 0.1$. A sensitivity filter with radius $r_{\min} = 1.5h$ (where $h$ is the element size) is applied to avoid checkerboard patterns and ensure mesh-independent solutions. The optimization runs for 51 iterations. This problem corresponds to a specific instantiation of the general objective $\Gamma(U, \rho)$ in Eq. 10, where $\Gamma$ is chosen as the structural compliance.

### B.4.2. RESULTS

Figure B.19 shows the boundary condition setup and the convergence history. The compliance decreases rapidly during the first 20 iterations and converges to a stable value, indicating successful optimization. The final compliance is reduced by approximately 36% compared to the initial uniform design while satisfying the volume constraint.

Figure B.20 illustrates the evolution of the optimized density field throughout the optimization process. Starting from a uniform density distribution ($\rho = 0.5$), the algorithm progressively develops a well-defined truss-like structure that efficiently transfers the applied load to the fixed boundary. The final design exhibits clear 0-1 material distribution with minimal intermediate densities, demonstrating effective penalization.

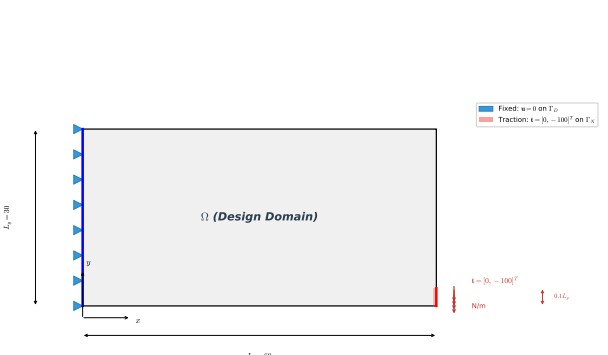

*(a)* Problem setup: fixed left boundary (blue) and applied traction load at bottom-right corner (red arrows).

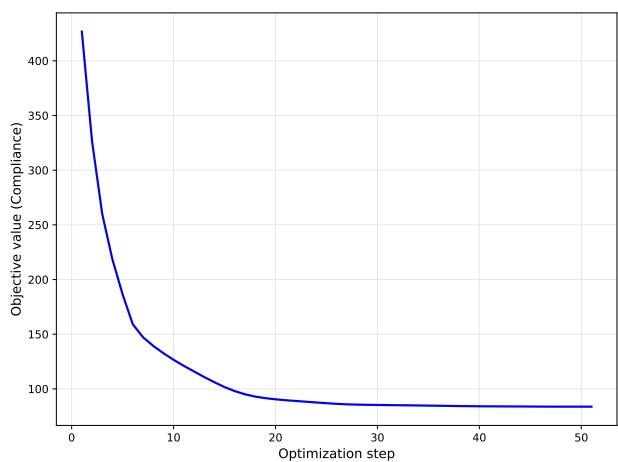

*(b)* Convergence history: compliance vs. optimization iteration.

*Figure B.19.* Topology optimization setup and convergence. (a) Boundary conditions for the cantilever beam problem. (b) The objective function (compliance) converges within 51 iterations.

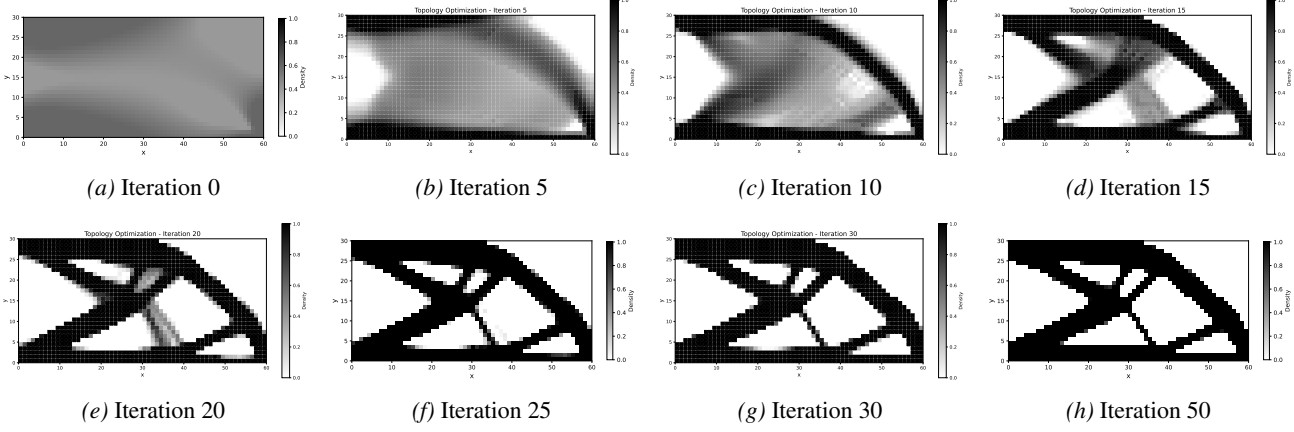

*(a)* Iteration 0   *(b)* Iteration 5   *(c)* Iteration 10   *(d)* Iteration 15

*(e)* Iteration 20   *(f)* Iteration 25   *(g)* Iteration 30   *(h)* Iteration 50

*Figure B.20.* Evolution of the optimized density field during topology optimization. The grayscale indicates material density ($\rho = 0$: void, $\rho = 1$: solid). The structure evolves from a uniform distribution to a well-defined truss-like topology.

