# OpenReview forum: "Learning, Solving and Optimizing PDEs with TensorGalerkin: an efficient high-performance Galerkin assembly algorithm"
_ICML.cc/2026/Conference — ICML 2026 regular_

### Official Review · Reviewer_M1Uj · 2026-03-03

**Soundness:** 3
**Presentation:** 2
**Significance:** 3
**Originality:** 3
**Overall Recommendation:** 5
**Confidence:** 5

**Summary:**

The paper proposes a unified algorithmic framework for the numerical solution, physics-informed learning, and constrained optimization of PDEs that possess a variational structure. To overcome the computational inefficiencies of standard FEM assembly in Python-based automatic differentiation frameworks, specifically the interpreter overhead and graph fragmentation caused by element-wise scatter-add loops, the authors reformulate the Galerkin assembly process into a strictly tensorized Map-Reduce operation. The initial stage computes local element matrices simultaneously via dense tensor contractions, successfully eliminating loops over elements, basis functions, and quadrature points. The subsequent stage aggregates these local contributions into a global sparse matrix using precomputed, topology-aware routing matrices and deterministic sparse matrix multiplication.This differentiable assembly engine is subsequently evaluated across three downstream tasks. First, it functions as a GPU-accelerated numerical PDE solver, with the authors reporting significant speedups over traditional CPU-based FEM libraries and a JAX-based differentiable solver on 3D benchmarks. Second, it is utilized for physics-informed operator learning; by evaluating discrete variational residuals using the assembly engine, spatial derivatives are computed analytically via shape function gradients, intentionally avoiding the deep computational graphs and AD overhead typical of strong-form methods like PINNs. Finally, the framework is applied to PDE-constrained inverse design, such as topology optimization, where gradients are seamlessly backpropagated directly through the sparse linear system and its assembly process without requiring separately formulated adjoint equations.

**Compliance With Llm Reviewing Policy:**

Affirmed.

**Final Justification:**

While the authors did initially overstate some of their novelty claims (as noted in my initial review), I believe the paper's core contribution to SciML remains significant once those claims are properly calibrated. The diverse, interdisciplinary pieces make it a substantial contribution. Therefore, I recommend to accept it.

**Key Questions For Authors:**

1. The paper states that all methods were restricted to the same network backbone and optimizer schedule. Can you provide an ablation or updated results where the baselines (PINN, VPINN, Deep Ritz) are properly hyperparameter-tuned (e.g., tailored learning rate schedules, boundary loss weighting, etc) to their respective convergence?
2. Figure 3 only shows the forward loss computation on regular grids. Can you provide the wall-clock scaling for the backward pass specifically on highly unstructured meshes, where SpMM on GPUs is known to suffer from uncoalesced memory access?
3. The scaling in Figure 3 stops at $10^7$ DOFs. Can you explicitly address the VRAM memory ceiling of your approach compared to traditional CPU-based FEM solvers, which routinely scale to hundreds of millions of DOFs using system RAM?
4. How does the "Sparse-Reduce" stage fundamentally differ from standard boolean routing and SpMM vectorization techniques already widely established in the HPC and FEM communities, as well as the DL/NLA intersection?

**Limitations:**

The authors have thoughtfully identified a core mathematical limitation of their work, specifically, that the framework currently requires the underlying PDE to possess a variational structure. However, there are a few computational and hardware constraints that would benefit from further discussion to provide a more complete picture of the method's applicability:
1. While the framework elegantly batches element-wise operations, its reliance on PyTorch's native automatic differentiation engine introduces a VRAM ceiling. Even though the local degrees of freedom ($k$) per element are few by definition, evaluating the physics across millions of elements simultaneously while retaining the intermediate buffers required for the autograd graph consumes significant GPU memory. As a result, the maximum DoFs may be constrained by VRAM (as evidenced by the scaling graph stopping at $10^7$ DoFs ). Acknowledging this memory limit, especially in contrast to traditional CPU-based FEM solvers that leverage abundant system RAM to scale to hundreds of millions of DoFs, would provide valuable context.
2. The scaling experiment for the forward loss computation (Figure 3) is explicitly performed on "regular grids". Because SpMM performance is highly sensitive to the uncoalesced memory access patterns typical of highly unstructured meshes, the idealized scaling shown may not fully translate to complex, real-world geometries. Showing behavior on more challenging unstructured domains would be helpful.
3. While the paper successfully demonstrates end-to-end differentiability and benchmarks the full optimization loop for an inverse design task (Table 3), this experiment is conducted on a very small 2D mesh (~2000 nodes). The authors do not provide scaling benchmarks for the backward pass as DoFs increase. Relying on PyTorch's native reverse-mode automatic differentiation to backpropagate through an iterative sparse linear solver (e.g., BiCGSTAB) typically introduces severe memory overhead and gradient instability. Given that the forward-pass scaling graph (Figure 3) stops at $10^7$ DoFs, it is highly likely the backward pass would hit a VRAM ceiling much earlier. The authors should explicitly discuss the technical and memory limitations of differentiating through the full solver pipeline at scale.

**Strengths And Weaknesses:**

## Strengths

* The paper presents a highly practical systems-level optimization by formulating the Galerkin assembly process completely within PyTorch. By decoupling the local physics evaluation from the global topological aggregation, the framework effectively bypasses the graph fragmentation and Python interpreter overhead that typically plague native AD-based FEM assembly.
* The purely tensorized assembly allows the framework to leverage PyTorch's native autograd engine directly through the sparse linear system. This provides an elegant, end-to-end differentiable pipeline for PDE-constrained inverse design without the need to derive and implement custom adjoint equations.

## Weaknesses:
* The paper heavily frames the "Sparse-Reduce" stage, i.e., using precomputed boolean routing matrices to map local contributions into a global CSR matrix via SpMM, as a novel algorithmic paradigm. However, transforming scatter-add assembly loops into sparse matrix-vector or matrix-matrix products is a well-established vectorization technique in the HPC and finite element communities. While the PyTorch implementation is highly useful, framing this standard HPC primitive as a core algorithmic breakthrough may be an overstatement for this ML venue.
* The authors claim significant computational efficiency gains over existing baselines. However, for the numerical solver, they predominantly compare against CPU-bound frameworks (FEniCSx, scikit-fem) and exactly one differentiable GPU solver (JAX-FEM). Beating legacy CPU libraries on a GPU is expected. The evaluation noticeably omits comparisons against a broader landscape of modern, highly optimized GPU simulation and differentiable physics frameworks. The paper also fails to situate itself within the broader field of differentiable/GPU-accelerated FEM, which already has a broad body of work over the last decade at the very least. Without these, the claim of state-of-the-art performance is empirically incomplete.
* Note that the FEM framework used is FEniCSx, i.e., Barrata et al. 2023, which is distinct from the older FEniCS library.
* In the Neural PDE Solver evaluation, the authors state, "We enforce a strict control setting: all methods use the same network backbone, mesh, and optimizer schedule". While intended to be fair, I find this to be a severe methodological flaw. Different learning paradigms (e.g., strong-form PINN residuals vs. energy minimization in Deep Ritz vs. discrete residual minimization in TENSORPILS) may possess drastically different optimization landscapes, conditioning, and spectral biases. Forcing all models to use an identical SIREN architecture and the exact same schedule of 10,000 Adam plus 200 L-BFGS steps, without individually tuning crucial hyperparameters like boundary condition loss weights or learning rate decay, may cause sub-optimal performance for the baselines. Consequently, the massive accuracy gaps reported (e.g., in Table 1) cannot be reliably attributed to the superiority of the proposed method versus poor baseline tuning.
* The scalability analysis is incomplete and may mask fundamental hardware limitations. First, while the authors claim optimization of the backward pass, Figure 3 only benchmarks the forward loss computation. Second, the evaluation in Figure 3 is explicitly performed on "regular grids". Regular grids produce highly structured, banded sparsity patterns that artificially inflate SpMM performance on GPUs by avoiding the uncoalesced memory access penalties inherent to unstructured meshes. Finally, the scaling graph abruptly stops at $10^7$ DOFs, presumably due to GPU VRAM limits. The authors do not acknowledge that while CPU-based FEM solvers can scale to massively larger industrial problems, GPU-based assembly frameworks often hit a hard memory wall, limiting their broader applicability.
* This reviewer finds the manuscript difficult to follow. It relies heavily on buzzwords and assumes deep familiarity with FEM jargon without providing sufficient explanations for a general ML audience. Furthermore, the text needlessly repeats information and includes material that has little bearing on the core contribution, struggling to just get to the point. Conversely, it omits critical methodological and experimental details, excessively offloading them to the Supplementary Material (i.e., constantly deferring to the appendices).

---

> ### Author Rebuttal · Authors · 2026-03-30
>
> We start by thanking the reviewer for carefully reading our paper, appreciating its contributions as **elegant, highy practical and highly useful**. We answer your questions below:
>
> 1. **W1/Q4**: The reviewer is correct in asserting that the Reduce with SpMM might be known in the HPC community although we would be grateful for a precise reference, we contend that SpMM is only one ingredient of our pipeline. As you astutely observed, our *systems-optimized pipeline* uses the Galerkin assembly algorithm for pruning the computational graph to O(1) nodes and unlocking downstream applications (Physics-informed learning, optimization etc) by making efficient back propagation possible. To the best of our knowledge, this overall pipeline is a novel contribution. Moreover, the assembly itself is a Map-Reduce operation where the crucial Map stage consists of substituting loops over elements, quadrature points and basis function into a single dense tensor via *einsum*. We find this element of our algorithm to be novel (as acknowledged by other reviewers) and also stress that the co-design of the Map-Reduce algorithm is an essential and novel feature (pl. also see Pt. 1 of our reply to Rev. isXW).
>
> 2. **W2/W3**: We compare our TensorMesh numerical PDE solver with very popular Fenics (in the SOTA Fenicsx version), SKFEM and GPU-compatible JAXFEM. Following your excellent suggestion and that of Rev. isXW, we have compared with 3 more baselines -- Firedrake (highly optimized SOTA PDE solver), MFEM and torch-fem (GPU compatible PDE solver). We request the reviewer to see Pt.  2 in our reply to Rev. isXW for more details on these solvers as well as our results which show that TensorMesh is significantly faster to run (at the same accuracy) than both CPU-based and GPU-based baselines. We believe that this GPU-GPU comparison on the same hardware is a fair evaluation of the methods.  Despite the perception, we did not find any significant GPU-compatible FEM solvers (other than JAX-FEM and torch-fem) and would be grateful to the reviewer for a precise reference. We would be happy to compare with any such solver in a camera ready version (CRV), if accepted.
>
> 3. **W4/Q1**: Following the reviewer's suggestion that the baselines in Tab. 1 be further hyper parameter tuned, we performed a per-method grid search on varying learning rates and boundary loss weights in the K=8 case resulting in Errors with PINN and VPINN being reduced to 8.57% and 7.79%, while the DeepRitz error remained at the reported value of 10.6%. To provide a fair comparison,  we also hyperparameter tuned TensorPILS by varying learning rates and Siren frequency (it has no boundary weights), resulting in the error reducing to 7.51% error. At this stage, we would really to emphasize that the *main advantage with our approach is not necessarily its accuracy but rather its speed*. These new results reinforce our point that even with comparable accuracy, TensorPILs continues to be significantly faster than baselines, i.e. 5x faster than PINN and 2x faster than VPINN and Deep Ritz.
>
> 4. **W5/Q2**: The reviewer's question on providing scalings for forward and backward pass with our algorithm and baselines (Fig. 3) on unstructured grids is very valid. We performed such an analysis on unstructured triangular grids and will report the corresponding figures in a CRV. Summarizing, for forward pass, TensorPILS is 3.5-5.8x faster than PINN as they require AD graph construction for computing 2nd derivatives, for backward pass, the performance gap is wider e,g, for 2.3M DoFs, PINN's backward pass takes 73.65ms whereas TensorPILS' takes 12.27 ms providing a 6x speedup and showing the advantage of our O(1)-sized computational graph in backprop. Finally, TensorPILS scales linearly with DoFs for both forward and backward pass (the ratio remaining constant), not suffering from any *uncoalesced memory access penalties* as the reviewer might have feared.
>
> 5. **W5/Q3**: in Fig. 3, we stopped the graph at around 6M DoFs as the baseline PINN ran out of memory. TensorPILs is more memory efficient and can be run on the same GPU for 12M DoFs. For larger problem sizes, we utilize the Multi-GPU parallel assembly of our framework and ran tests on 4-GPU RTX6000-pro to find that TensorPILs has perfect linear scaling in memory and we can run upto 110M DoFs on this system. This linear scaling also entails that there is no inherent VRAM ceiling and larger problem sizes can be readily fit into larger number of GPUs.
>
> 6. **L3**. We clarify a possible misunderstanding: solver gradients for topology optimization use the adjoint method (one extra linear solve, O(1) graph), not naive AD through BiCGSTAB, with no memory overhead, removing any concerns on memory limitations in this regard.
>
> We hope that we have addressed the reviewer's concerns, particularly on novelty, baselines and scaling/memory ceiling, to your satisfaction. We kindly request the reviewer to upgrade your assessment accordingly.

---

> > ### Author Rebuttal · Reviewer_M1Uj · 2026-03-31
> >
> > 1. Formulating a reduction operation as an SpMM or expressing data routing, scatter/gather, and parallel reduction operations as sparse matrix multiplications is a known technique in HPC and numerical linear algebra. For example, Algebraic Multigrid methods have historically utilized sparse restriction matrices to perform weighted reductions of dofs that go back at least to Ruge & Stüben, 1987. Similarly, in graph processing, frameworks like Combinatorial BLAS (Buluç & Gilbert, 2011) explicitly define neighborhood data reductions as SpMV/SpMM operations over boolean incidence matrices.
> > While I do agree that the overall pipeline and implementation are novel and very useful, this specific claim needs to be removed from the paper. It is sufficient to state that the authors used this technique to enable the true novel contributions of the paper.
> > 2. While there are other GPU-based FEM assemblers, I find that the new reported evaluations are sufficient to make a compelling case for TensorGalerkin. I stress again that FEniCSx is distinct from the original FEniCS, created about 20 years after the original by a different set of authors. The citation in the paper therefore should be Barratta et al. 2023 instead (or in addition to) the original FEniCS paper.
> > 3-4. I find these evaluations sufficient and request that they be included in the CRV along with a description of the unstructured grids and tuning methods used.
> > 5. Using just 4 GPUs is not enough to claim "perfect linear scaling" but I appreciate the experiment and believe this proves the utility of the method for much larger grids.
> > 6. My concerns regarding the writing have not been addressed. I believe this is a good paper with a concrete novel contribution, but is quite hard to read, especially for a general ML audience, and this matters. Please commit to improving the writing so that this interesting paper can be read by as many ICML readers as possible.
> >
> > I thank the authors for their efforts in the creation of the paper and effort expended for the rebuttal, and wish them all the best. Good luck.

---

> > > ### Author Response · Authors · 2026-04-01
> > >
> > > We sincerely thank the reviewer for their prompt reply and positive response to our rebuttal. We answer their follow-up questions below while also acknowledging that this reply provides us with an opportunity to answer some of the questions raised in the original review that we could not discuss in our rebuttal due to space limitations.
> > >
> > > **P1**: We are very grateful to the reviewer for pointing us to the specific references in the HPC and Graph processing literature which clearly discuss the use of SpMM. Following your excellent suggestion, we will reframe our contribution in a CRV (if accepted) by clearly stating in our description of the Sparse-Reduce stage of our algorithm that *SpMM-based reduction is a well-established technique in HPC and graph processing (cf. Ruge and Stüben, 1987 and Buluç and Gilbert, 2011), and that our contribution in this context, lies in integrating this technique within a co-designed Map-Reduce pipeline in PyTorch's Automatic Differentiation framework, achieving O(1) computational graph complexity for an efficient end-to-end differentiable Galerkin assembly*. We hope that this framing assuages this valid concern of the reviewer.
> > >
> > > **P2**: The reviewer is absolutely correct in pointing out that FEniCSx is indeed the latest version of the FEniCS framework and we will acknowledge it as such in a CRV. Inspired by this point of the reviewer, we ran experiments with this highly optimized 8-Rank CPU version of FeNICSx for the 3D Poisson and Linear Elasticity examples to obtain the following results: i) For 3D Poisson at 741K DoFs, TensorMesh continues to be the fastest solver achieving a speedup of 1.52x over FeNICSx and 1.7x over FireDrake and ii) for 3D Elasticity at 91K DoF, TensorMesh performs even better relatively by achieving speedups of 7.4x over FeNICSx and 10.7x over FireDrake. These additional experiments confirm that TensorMesh performs very well as a fast numerical PDE solver, compared to the state of the art. Needless to say, we will add the FeNICSx and FireDrake results to a modified version of Fig. 2 in a CRV, while also emphasizing that these frameworks do not directly allow for Physics-informed operator learning or PDE constrained optimization.
> > >
> > > **P3**:  We agree with the reviewer that the observation of linear scaling upto 4 GPUs is not a definitive demonstration of scaling TensorGalerkin to a large number of GPUs and will frame this result as an indicator of the distributed computing capacity of our framework. We aim to perform scaling experiments for larger number of GPUs and include them in a CRV, if accepted.
> > >
> > > **P4**: This was one of the points that we could not address in our original rebuttal due to space limitations. The reviewer's point on the readability of our paper is very well-taken. We are committed to improving the paper's readability and consequently, its reach to the broader ML community. Aided by the fact that the CRV admits an extra page, we will strive to explain some of the jargon that has been used and put the contributions into proper perspective.
> > >
> > > We thank the reviewer again for their constructive criticism which will enable us to improve the presentation and contents of our paper.

---

### Official Review · Reviewer_SXpy · 2026-03-05

**Soundness:** 3
**Presentation:** 3
**Significance:** 2
**Originality:** 3
**Overall Recommendation:** 4
**Confidence:** 4

**Summary:**

This paper proposes a high-performance AD/GPU-compliant TensorGalerkin framework for assembling Galerkin finite element operators for ML environments. TensorGalerkin employs a two-stage process for kernel assembly and enables three downstream applications within a unified PyTorch pipeline. Experiments on several PDE benchmarks show significant speedups compared to SOTA while maintaining similar accuracy. Although this work is mainly an engineering effort instead of a technical innovation, the empirical speedups are promising enough that I'd recommend acceptance.

**Compliance With Llm Reviewing Policy:**

Affirmed.

**Key Questions For Authors:**

- What iterative solvers and preconditioners are used in experiments? For solvers, these are equally important as assemblers.
- The header reads as "Submission and Formatting Instructions for ICML 2026”. This should be replaced with the actual title.

**Limitations:**

- Yes.

**Strengths And Weaknesses:**

- **Strengths**
  - Replacing scatter-add loops with tensor contractions and SpMM is a novel innovation.
  - Identification of computation bottlenecks in existing FEM/AD pipelines and the efficient speedups.
  - Unified framework for multiple downstream applications and workflows.
  - Strong experimental results with significant efficiency gains.
- **Weaknesses**
  - This work lies primarily in systems engineering rather than algorithmic advances in AI for math.
  - The experiments are on linear, time-independent PDEs. Generalization remains a question.
  - Although assembly speed is emphasized, the paper gives relatively little attention to iterative solver performance.

---

> ### Author Rebuttal · Authors · 2026-03-30
>
> We thank the reviewer for your careful reading and appreciating the novelty and contributions of our paper. We answer your questions in detail below:
>
> **W1**: Regarding the reviewer's concern about the fact that the work *lies primarily in systems engineering* rather than *algorithmic advances in AI for Math*, we would like to answer by elaborating on the following algorithmic/methodological novelties of our approach. As astutely acknowledged by the reviewer, we provide a **highly efficient end-to-end fully differentiable algorithmic pipeline** that acts as a i) numerical PDE (FEM) solver ii) neural PDE solver iii) learns solution operators of PDEs in a physics-informed manner without needing any training data and iv) its full pipeline differentiability enables numerous downstream tasks such as PDE-constrained optimization and Inverse problems. Thus, our contributions should be judged in each of these categories:
>
> i) **Physics-Informed Operator Learning**: Compared to data-driven approaches, Physics-informed operator learning, dominated by PINNs and their extensions such as physics-informed neural operators is plagued by the inefficiency of the *graph-within-a-graph* bottleneck in automatic differentiation, leading to very poor performance on operator learning tasks when compared to purely data-driven approaches, To ameliorate this bottleneck, we proposed a fundamentally different approach by analytically computing shape gradients within a Galerkin framework, while ensuring a O(1) computational graph that is optimal for backpropagation. We consider this as an algorithmic novelty rather than just systems engineering and its utility is demonstrated by the results for operator learning tasks reported in Tab. 3 where we significantly outperform both data-driven and physics-informed baselines.
>
> ii) **Impact on AI for Science**: We believe that our contribution in this regard is two-fold. In addition to the physics-informed learning and PDE constrained optimization aspects of our framework, we would like to emphasize the fact that the lack of efficient GPU-compatible FEM-type PDE solvers is an essential bottleneck for ML researchers working on AI for PDEs. By providing such a framework here, we contribute a solver for highly efficient *batched data generation* which is essential for creating large-scale datasets for PDE foundation models. Moreover, we can incorporate the latest hardware kernel optimizations and integrate seamlessly with ML models to enable hybrid learning at scale. Our goal is to enable new research directions in this field by presenting our end-to-end PDE solving and learning pipeline.
>
> iii) **Algorithmic novelty of our Assembly Algorithm**. We would like to succinctly summarize the novel aspects of our Map-Reduce assembly algorithm by stating that "Unlike prior approaches which have relied on compilers (JIT/XLZ/TSFC) to fuse fragmented element-wise operations at runtime, we perform a novel *mathematical reformulation* that algebraically compresses the 0($k^2$E) operations (k being local dofs and E number of element) into a very small (O(1)) number of large-scale tensor operations (einsum, SpMM), achieving significant graph compression which is the key to efficient back propagation, enabling further downstream tasks. Moreover, these gains accrue to us while still handily outperforming widely popular frameworks such as Fenics and Firedrake on numerical PDE solution tasks (please see reply to Rev. isXW, pt. 2).
>
> We hope that the cumulative contributions are viewed in terms of algorithmic novelty and not just systems engineering.
>
> **W2**: We have already presented results on both **time-dependent as well as nonlinear PDEs** in our original version, please check from Section 3 and results from Table 2, where we have considered operator learning tasks for the linear time-dependent Wave equation (prototypical hyperbolic PDE) as well as the non-linear Allen-Cahn equation (prototypical nonlinear parabolic PDE). For both cases, we document performance on in-distribution and out-of-distribution learning tasks to find from Tab. 2 that our TensorPILS framework is more accurate than the data-driven framework, while being an order of magnitude more accurate and generalizable than the data-free Physics-informed DeepONet baseline. Further details on the use of our algorithms for nonlinear PDEs is provided in SM A.1 (L605-633)
>
> **W3**. The iterative solver that we use is presented in SM Tab. B.1 (L770-777). As our focus in this paper was on ameliorating the assembly bottleneck on modern GPUs, we used the existing fairly mature iterative solvers (even on GPUs) for solving the underlying linear equations and did not further pursue this topic here.
>
> We hope that we have addressed the reviewer's concerns, particularly with respect to algorithmic novelty of our pipeline and time-dependent and nonlinear PDE benchmarks, to your satisfaction. We request the reviewer to kindly upgrade their assessment accordingly.

---

> > ### Author Rebuttal · Reviewer_SXpy · 2026-04-01
> >
> > I'd like to thank the authors for their rebuttal, as that better explains the content of the paper. However, I keep my original view that this paper is engineering-based. The "Map-Reduce assembly algorithm" is migrating an existing idea to a "new" domain, and this is an engineering adaptation, not a theoretical/algorithmic breakthrough. I'm keeping my original rating.

---

> > > ### Author Response · Authors · 2026-04-01
> > >
> > > We thank the reviewer for their prompt response. While we strived to highlight the algorithmic innovations, we respect the reviewer's perspective. We would like to emphasize that this *engineering adaptation* was necessary to unlock downstream applications such as numerical and neural PDE solving, Physics-informed operator learning and PDE constrained optimization, which were not possible within a single end-to-end pipeline before, to the best of our knowledge.

---

### Official Review · Reviewer_isXW · 2026-03-12

**Soundness:** 3
**Presentation:** 4
**Significance:** 4
**Originality:** 3
**Overall Recommendation:** 4
**Confidence:** 4

**Summary:**

The paper presents a framework for the matrix assembly problem for Galerkin type approximations of time-dependent PDE solutions and for operator learning in the context of PDEs. The introduced TensorGalerkin framework performs operator learning by solving PDEs with a Galerkin-based discretization and standard iterative solvers, while optimizing for the representation of the operator using gradient-descent-based methods. The framework is evaluated in comparison to other neural-network-based operator learning methods and to a Python-based PDE solver.

**Compliance With Llm Reviewing Policy:**

Affirmed.

**Final Justification:**

The rebuttal and discussion provided some justification and clarification of the core contributions, and the proposed revisions for the final draft sound reasonable.

**Key Questions For Authors:**

Please address issues raised in weaknesses above / clarify contributions.

**Limitations:**

yes

**Strengths And Weaknesses:**

Strengths:
 * The experimental evaluation appears to be extensive, considering a variety of settings and metrics, including both standard PDE solves and operator learning. Multiple baselines are also considered.
 * The presentation of the paper is generally easy to follow.
 * The paper appears to make contributions both on the side of operator learning and the lower-level problem of Galerkin matrix assembly.

Weaknesses:
 * The main contribution of the paper is not clear to me, is the operator learning procedure novel (I guess not so much)? Or is the main contribution the Galerkin matrix assembly? If its the latter, the related work discussion and experimental comparisons are not really adequate, they should consider direct comparison to prior work, such as Brenner and Scott, whose works appear to be leveraged, as well as modern high-performance FEM matrix assembly frameworks, most notably (to the best of my knowledge), FiredRake.
 * The concrete novelty and benefit of the design decisions made in the matrix assembly are not clear to me, is leveraging SpMM novel? How does the SpMM algorithm used compare to literature? The map-reduce framework for communication/parallelization is relatively naive, a computational/communication cost analysis would be more convincing in justifying the proposed method. I believe there are recent work that provide efficient algorithms for FEM matrix assembly and quantify the cost in detail (up to constants).

Overall, I liked the approach in the paper, and the experimental evaluation is well above the mark in level of detail in comparison to typical papers published in machine learning conferences. However, I think there are some critical comparisons missing, I would be supportive of publication of the paper at ICML if a comparison of the FEM assembly algorithm to prior works was added, as well as an experimental comparison to FiredRake or another high-performance FEM matrix assembly framework (or perhaps if the authors conv
~

---

> ### Author Rebuttal · Authors · 2026-03-30
>
> We thank the reviewer for their careful reading of our paper and appreciating our contributions as well as constructive criticism and questions which we answer below.
>
> 1. **W1**: In our opinion, our paper has *several* major contributions. As acknowledged by other reviewers, we present an **highly efficient end-to-end fully differentiable algorithmic pipeline** that acts as a i) numerical PDE (FEM) solver ii) neural PDE solver iii) learns solution operators of PDEs in a physics-informed manner without needing *any* training data and iv) full pipeline differentiability enables numerous downstream tasks such as PDE-constrained optimization and Inverse problems. As we wrote, the key to unlocking the downstream tasks of solving, learning and optimizing PDEs lies in the efficient matrix assembly, which was the major bottleneck in realizing an efficient algorithmic framework in this context. In terms of novelty of the proposed matrix assembly algorithm, we identify  i) it is designed to eliminate all loops over elements, basis functions and quadrature points by lifting them to a dense tensor using *einsum*. ii) ensuring that our algorithm yields O(1) nodes in the computational graph rather than a O($k^2E$) (k local dofs each over E elements) graph from standard AD approaches such as PINNs and their variants. This is the key to efficient back propagation and allows us to train neural PDE solvers/operators in a data-free physics-informed manner and iii) both above steps are *co-designed* to ensure end-to-end differentiability for additional downstream tasks. All these novel steps have to be carefully designed together, resulting in a highly efficient pipeline that enables learning, solving and optimizing PDEs on modern hardware at scale. We believe that these innovations constitute significant algorithmic novelty and will sharpen our contributions in a camera-ready version (CRV), if accepted.
>
> 2. **W1 (Further Baselines)** In our paper, we had already compared with 3 numerical PDE solvers (Fenics, SKFEM and JAX-FEM). Following the reviewer's excellent suggestion, we compare with 3 more: Firedrake (as suggested by you is a highly CPU & multi-rank-MPI framework optimized with UFL/TSFC/PyOP2), MFEM and the CUDA-based torch-fem. We tested these additional frameworks on the 3D Poisson and elasticity numerical PDE solving benchmarks (Fig.2) to observe that i) on Poisson with 741K DoFs, our TensorMesh provides significant speed-ups over Firedrake (1.7x), MFEM (25x) and torch-fem (2.1x) and ii) these speedups are even larger for the elasticity benchmark with 91K DoFs as TensorMesh is much faster than Firedrake (10.7x), MFEM (170x) and torch-fem (20x).  These results supplement the observations from Fig.2 where we showed even larger speedups with respect to Fenics, SKFEM and JAX-FEM frameworks. At the same time, we need to emphasize that Firedrake, MFEM and torch-fem cannot be used for either PDE operator learning or optimization. We will include these results in a CRV to further strengthen the empirical basis of our paper.
>
> 3. **W2**: We politely disagree with the statement on the naivety of our *Map-Reduce* algorithm as i) the map stage is designed to replace all loops (over E elements, Q quadrature points and $k^2$ basis function pairs) via an *einsum* to yield a dense [E,k,k] tensor ii) which automatically fits into the SpMM Reduce Stage, without format conversion, where the precomputed routing matrices decouple mesh topology from physics and iii) Map-Reduce are co-designed to ensure a computational graph with O(1) nodes which is essential for efficient back propagation needed for our downstream operator learning tasks. Each cog in this pipeline has been designed to fit seamlessly such that the end-to-end pipeline remains efficient. We can show that the map-reduce stage yields the optimal assembly compute complexity of O(EQk^2) flops but the algorithm does much more in the form of i) near-peak GPU utilization due to dense tensor contractions ii) minimal autograd overhead for physics-informed learning and iii) no JIT recompilation (in contrast to JAX-FEM) as it only requires one-time cost of recomputing sparse matrices for Reduce. To the best of our knowledge, we have not come across any paper in the ML or HPC literature where all the afore-mentioned algorthmic elements have been combined in a single end-to-end PDE solving, learning and optimizing pipeline and would definitely compare with these references if the reviewer points to them. Needless to say, we plan to add further extensive discussions in the CRV on the algorithmic novelty of our entire framework and the assembly algorithm.
>
> We hope that we have addressed the reviewer's concerns, particularly with respect to algorithmic novelty of our pipeline and of our matrix assembly as well as the comparison with Firedrake to your satisfaction. We request the reviewer to kindly upgrade their assessment accordingly.

---

> > ### Author Rebuttal · Reviewer_isXW · 2026-04-05
> >
> > Thank you for the response. The new compairson to FiredRake and other framework sounds good. However, after reading the other reviews and the responses and looking at the paper again, I think I my point of view on the contributions is more negative than before. Throughout your responses, you point out to different elements of the paper that are novel and also highlight the end-to-end nature. Reading through this, its not clear to me that I see any single clear significant novel contribution, and I have skepticism regarding a number of them. I would rather see a small number of well-justified and deep contributions than a laundry list of things that have questionable value/novelty/impact. In particular, (1) I had thought on my first read that the operator learning method you are using is derived from prior work. If the mathematical formalism used for it is novel, I think there needs to be quite a bit more comparison to existing methods, and discussion of the mathematical relationship to other method. The paper flows from background on Galerkin into stating what is apparently a new method too quickly. From a quick view of related work, papers like "Li, Zongyi, et al. "Physics-informed neural operator for learning partial differential equations." ACM/IMS Journal of Data Science 1.3 (2024): 1-27." then need to be discussed in detail, currently the paper cites this but dismisses the work as impractical due to use of automatic differentiation, which I don't find to be an adequate point of criticism. (2) In some of your responses, you point out novelty of the implementation in (1) use of SpMM, (2) embedding the FEM computation into an einsum. Both are relatively simple matters of representation, the latter I think is well-understood at the element level and trivial across elements (the use of batched GEMM for this is well-studied), SpMM is also frequently employed in FEM. Moreover, none of these components/novelties are really studied specifically and contrasted to related work at the level of algorithmic details. I also did now find the justification of map-reduce particularly useful, I think you should compare to existing parallelization strategies and alternatives quantitatively (communication cost, etc.) if you want to justify this. (3) I think there are many "end-to-end" papers on numerical PDE solvers and operator learning approaches. This in itself is not a novelty, and mixing contributions that span the spectrum from low-level HPC (map-reduce parallelization) to method development (new operator learning algorithms) in one paper is confusing unless they are well coupled.

---

> > > ### Author Response · Authors · 2026-04-05
> > >
> > > We thank the reviewer for their comments and provide a detailed response to their points of concern below.
> > >
> > > **P1**. We start by reiterating the very rationale of our paper. *It is not just to provide a fast GPU-compatible numerical FEM solver but to unlock key downstream applications in the form of physics-informed operator learning and end-to-end differentiability which enables PDE-constrained optimization and inverse problems*. To achieve this goal, the core contribution of our paper is the TensorGalerkin assembly algorithm that reduces the underlying autograd computational graph from O(Ek²) nodes (for naive algorithms) to the optimal O(1) nodes (as in our method) for Galerkin weak-form assembly within PyTorch. This is clearly not a claim about our original contribution to einsum or SpMM individually. Rather, as the reviewer M1Uj has correctly noted (please see our extended discussion with Reviewer M1Uj below for further details on novelty as this reviewer had flagged concerns similar to yours), the main contribution is **the specific co-design that yields the O(1)-node property for the computational graph**, which is what makes physics-informed Galerkin operator learning and differentiable PDE-constrained optimization computationally efficient on GPUs. Solving, learning, and optimization are downstream applications that validate this assembly algorithm and should not be seen as independent contributions by themselves. We will restructure the CRV, if accepted, to make this hierarchy explicit and frame our contributions more sharply.
> > >
> > > **P2**. About PINO, we agree that our current discussion of PINO is rather condensed, due to space limitations. Comparing PINO with TensorPILs, we would like to point out that PINO enforces PDE constraints via strong-form residuals evaluated pointwise at collocation points in the domain, computing spatial derivatives through Automatic Differentiation (AD). On the other hand, our TensorPILS enforces PDE constraints via weak-form (Galerkin) residuals, computing spatial derivatives analytically through pre-computed shape function gradients. The key difference is not merely "AD vs. no AD" — it also encompasses the fact that weak-form discretization (i) naturally handles low-regularity solutions and complex geometries where strong-form residuals are ill-defined, and (ii) when combined with our O(1)-graph assembly, avoids the deep computational graphs that arise from differentiating through loops, impinging on the efficiency of AD-based methods. Furthermore, *PINO is restricted to Cartesian grids with the FNO as the backbone, but our framework handles arbitrary unstructured meshes*. The two benchmarks for operator learning that we studied in our paper involve solution operators of PDEs on non-Cartesian domains making it impossible for us to compare with PINO. Rather, we compared with Physics-informed (PI) DeepOnets, a very similar framework as PINO in terms of strong-form residual computation with AD, but with a DeepONet backbone replacing FNO, that allows for operation on non-Cartesian domains. As shown in Tab. 2 and Figs B.13,14,15, TensorPILs is 10x more accurate than PI-DeepONet for these benchmarks.  We will add a detailed discussion, comparing TensorPILs with PINO in a CRV, if accepted.
> > >
> > > **P3** On “end-to-end” novelty, the reviewer states that there are "many end-to-end papers on numerical PDE solvers and operator learning." To the best of our knowledge, this is *not* the case and we are happy to concede this point if a suitable reference is provided. In our opinion, the most suitable baseline that supports both PDE solving and PDE-constrained optimization in one package is JAX-FEM. We have compared against JAX-FEM extensively in our paper: for numerical PDE solving TensorMesh achieves order-of-magnitude speedups over JAX-FEM on the same GPU hardware; for PDE-constrained optimization, TensorOpt achieves a 4× speedup over JAX-FEM on the topology optimization benchmark while converging to essentially identical designs. Critically, JAX-FEM does not support data-free physics-informed operator learning or neural pde solving — capabilities uniquely enabled by our O(1)-computational graph assembly.
> > >
> > > To summarize, while conceding (as we have done multiple times in our paper) that there are plenty of frameworks which individually target numerical PDE solving on MPI multi-rank CPUs (FeNICSx,FireDrake) and GPUs (JAX-FEM, torch-fem) or neural PDE solving (PINNs, Deep-Ritz) or Physics-informed Operator Learning (PINO, PI-DeepONet) or PDE constrained optimization (JAX-FEM), there is no single framework other than the one that we propose that can perform all these tasks, while at the same time significantly outperforming the afore-mentioned task-specific frameworks on each of these tasks.
> > >
> > > Given this additional information, we hope that the reviewer will appreciate this key contribution of our paper and adjust their assessment accordingly.

---

### Official Review · Reviewer_QFSy · 2026-03-15

**Soundness:** 2
**Presentation:** 2
**Significance:** 2
**Originality:** 2
**Overall Recommendation:** 2
**Confidence:** 4

**Summary:**

The work implements a GPU-based Galerkin-type FEM discretization, aimed at improving the speed and accuracy of PDE solvers. The proposed approach reduces the bottlenecks faced by traditional FEM and physics-informed operator learning automatic differentiation through a Map-Reduce framework. The contributions of this work is a Python-native, GPU-compatible FEM solver.

**Compliance With Llm Reviewing Policy:**

Affirmed.

**Final Justification:**

Thank you for the authors’ response. After careful consideration and multiple re-readings of the manuscript, I still have several concerns regarding the level of contribution and the overall presentation.

**Novelty** The core approach, including batched tensor computation for local assembly and pre-allocated sparse matrix construction, appears to follow standard vectorization practices and common strategies for sparse matrix assembly.

**Engineering depth** The current implementation mainly relies on high-level tensor operations on a single GPU. Demonstrating deeper system-level contributions, such as custom GPU kernels, memory optimization, or multi-GPU scalability, could significantly enhance the engineering impact.

**Writing**  The manuscript places substantial emphasis on downstream applications, such as PINNs and neural operators, which makes the main contribution on matrix assembly less clear and somewhat confusing.

**Impact** The method is primarily applicable to FEM-style formulations that explicitly construct matrices. Many modern PDE learning approaches, including neural operators and PINNs without explicit discretization, do not rely on such matrix assembly, which limits the broader applicability of the method.

**Key Questions For Authors:**

- Is there other related decomposition methods, and how it works (efficiency and accuracy) using the same GPU-implementation?
- Could you provide some example applications under other types of boundary conditions?
- Can you provide a discussion on accuracy improvement? For instance, which specific components contribute to the enhancement of accuracy?

**Limitations:**

yes

**Strengths And Weaknesses:**

**Strengths:**

- A Map-Reduce perspective is proposed for handling Galerkin assembly, implementing FEM on Python library GPU kernels, and using analytic shape gradients instead of automatic differentiation.
- Demonstrated improvements in both efficiency and accuracy on multiple benchmarks, including 2D and 3D elliptic, parabolic, and hyperbolic PDEs on unstructured meshes.

**Weaknesses:**

- Novelty: The major performanc gain comes from the new GPU-kernel implmentation of  operaterations, e.g. matrix multiplications. The author does not show novelty on the algorithm or new design.
- Lack of comparison of other methods. The manuscript does not provided any numerical comparison on other relavent decomposition  methods.
- The experiments on various PDEs only include cases with Dirichlet boundary conditions, and no cases with other types of boundary conditions are presented.

---

> ### Author Rebuttal · Authors · 2026-03-30
>
> We thank the reviewer for their careful reading of our paper and their constructive criticism and questions which we answer in detail below,
>
> 1. The reviewer summarizes our work as a contribution to *Python Native GPU-compatible FEM Solvers*. However, this is only one of our many contributions. As acknowledged by other reviewers, we present an **highly efficient end-to-end fully differentiable algorithmic pipeline** that acts as i) numerical PDE (FEM) solver ii) neural PDE solver iii) learns solution operators of PDEs in a physics-informed manner without needing *any* training data and iv) full pipeline differentiability unlocks numerous downstream tasks such as PDE-constrained optimization and Inverse problems. These contributions need to be viewed in their totality in our opinion.
>
> 2. **W1**:  Contrary to the reviewer's perception that the novelty in our work only lies in the GPU-kernel implementation of operations, we would like to emphasize that our novelty spans the following factors: i) the algorithm is designed to eliminate all loops over elements, basis functions and quadrature points by lifting them to a dense tensor rather than merely implementing GPU kernels for each operation. This step is the key to efficient assembly. ii) ensuring that our algorithm yields O(1) nodes in the computational graph rather than a O($k^2E$) (k local dofs each over E elements) graph from standard AD approaches such as PINNs and their variants. This is the key to efficient back propagation and allows us to train neural PDE solvers/operators in a data-free physics-informed manner and iii) both above steps are *co-designed* to ensure end-to-end differentiability for additional downstream tasks. All these novel steps have to be carefully designed together, resulting in a highly efficient pipeline that enables learning, solving and optimizing PDEs on modern hardware at scale. We believe that these innovations consistute significant algorithmic novelty and will sharpen our contributions in a camera-ready version (CRV), if accepted.
>
> 3. **W2/Q1**: Regarding comparison to other methods, we would like to point out that we had already compared with 3 state of the art Numerical PDE solvers (Fenics, SKFEM and JAX-FEM) in Fig.2, 3 neural PDE solvers (PINN, VPINN and Deep Ritz) in Tab. 1, 2 operator learning methods (fully data-driven and Physics-informed DeepONets) in Tab. 2 and 1 end-to-end differentiable pipeline (JAX-FEM) for PDE constrained-optimization in Tab. 3, constituting a significant effort compared to ML literature as acknowledged by  other reviewers. We have provided further comparisons (Pl. see Pt.2 in our reply to Rev. isXW) with 3 additional PDE solver frameworks (Firedrake, MFEM and torch-fem) to find that our TensorMesh handily outperforms them (faster by 10x or more).  We believe that these additional comparisons add to our narrative and will include them in a CRV.
>
> 4. **W3/Q2**: We had already provided an example with Mixed BCs in our paper in the topology optimization experiment (Sec. 3 and Tab. 3) where as described in SM B4 (l1312-1319, Eqs. B24 and B25), we impose a Dirichlet BC on one boundary segment and Neumann BC (in terms of traction load) on another. Nevertheless, following the reviewer's excellent suggestion, we have performed an additional experiment with even more complex BCs. To this end, we consider the setup of the recent paper (arXiv:2602.04923 ) for the Poisson Equation with 2 geometries (Circle and Boomerang Fig C.1) with Mixed Dirichlet, Neumann and Robin BCs (Sec C1.2) and solve the PDE with these complex BCs with our TensorMesh and Fenics, which was used in the underlying paper as the ground truth data generator. We observe excellent agreement of TensorMesh with Fenics (with a relative error $<10^{-4}$). On the other hand, TensorMesh is significantly faster than Fenics for both problems with speedups of 52x for Circle and 18x for Boomerang, showcasing the ability of our framework to handle extremely complicated BCs. We will present the results in a CRV and thank the reviewer for this suggestion that strengthens our work further.
>
> 4. **Q4**: Our main focus is on increasing speed although accuracy improvements over baselines are observed for physics-informed operator learning (Tab. 3), where we believe that computing shape gradients analytically and imposing BCs in a natural manner provides a more amenable loss landscape than other physics-informed neural operators. We plan to add a detailed discussion in a CRV.
>
> Summarizing, we re-emphasize that our novelty lies in providing i) highly efficient GPU assembly ii) highly optimized computational graph for back propagation iii) dynamic meshing without recompilation and iv) end-to-end differentiability which unlocks numerous downstream applications. We hope that we have addressed the reviewer's concerns, particularly about novelty, baselines and BCs, to their satisfaction and kindly request them to upgrade their assessment accordingly.

---

> > ### Author Rebuttal · Reviewer_QFSy · 2026-04-06
> >
> > Thanks for the authors' response. I appreciate the proposed method improve the speed of various PDE solvers, including traditional FEMs as well as PINNs, and neural operators.
> > After read the explanation and re-read the paper, the major contribution is still not clear, and I believe the manuscript needs a major revision for publication. Especially, the authors should clarify what algorithmic contributions regardless of hardware implementation?

---

> > > ### Author Response · Authors · 2026-04-06
> > >
> > > We thank the reviewer for their reply as this provides us with a further opportunity to address their concerns about the key contribution of our paper. Although we had explained our contributions in our original rebuttal, we are happy to elaborate on them further below.
> > >
> > > We start by reiterating the very rationale for our paper. To begin with, efficient numerical FEM solvers for PDEs on multi-rank CPUs (FeNICSx, FireDrake) and GPUs (JAX-FEM) are available. Some of them like JAX-FEM are also end-to-end differentiable and enable PDE constrained optimization. Moreover, there are also neural PDE solvers such as PINNs and Deep-Ritz although they are not efficient due to the bottlenecks caused by Automatic Differentiation (AD). Finally, there are physics-informed neural operators (Physics-informed DeepONets, PINO), which are also inefficient due to AD bottlenecks while computing spatial derivatives. However, to the best of our knowledge, there is no single unified framework that allows for PDE solving, learning and optimization within the same pipeline and does so efficiently on modern hardware. Why is this not the case ?
> > >
> > > We answered this question in our paper by identifying that the key obstruction for integrating numerical PDE solving and optimization with physics-informed operator learning is the standard realization of the FEM assembly algorithm, which led to the underlying computational graph containing O($k^2E$) nodes, with E elements and k basis functions per element. This deep computational graph leads to severe inefficiencies during automatic differentiation as one has to differentiate through loops, making physics-informed operator learning and PDE constrained optimization extremely inefficient.
> > >
> > > With this context, the key question that we wanted to answer is: can one design a novel algorithm for FEM assembly that leads to a much smaller O(1) computational graph ? Hence, our main contribution in the paper is precisely **the specific co-design of our FEM assembly algorithm (TensorGalerkin) that yields the O(1)-node property for the computational graph**, which is what makes physics-informed Galerkin operator learning and differentiable PDE-constrained optimization computationally efficient on GPUs. Solving, learning, and optimization are downstream applications that validate this assembly algorithm and should not be seen as independent contributions by themselves.
> > >
> > > We have also provided a very extensive discussion on this issue with other reviewers, especially Reviewer M1UJ and request the reviewer to kindly read this discussion below where further details are provided. As other reviewers have acknowledged, the TensorGalerkin algorithm is what unlocks the downstream applications and should be viewed as our main contribution.
> > >
> > > Given that the reviewer's main concern is in the proper and sharper framing of our contributions, rather than in any methodological or technical content, we politely disagree with the reviewer's assessment that the manuscript need a major revision. Rather, we will rewrite the introduction and conclusion of a camera-ready version (CRV), if accepted, to more sharply delineate our key contribution so that the readers can readily appreciate it.
> > >
> > > We summarize by repeating (as we have done multiple times in our paper) that although there are plenty of frameworks which individually target numerical PDE solving on MPI multi-rank CPUs (FeNICSx,FireDrake) and GPUs (JAX-FEM, torch-fem) or neural PDE solving (PINNs, Deep-Ritz) or Physics-informed Operator Learning (PINO, PI-DeepONet) or PDE constrained optimization (JAX-FEM), there is no single framework other than the one that we propose here that can perform all these tasks, while at the same time significantly outperforming the afore-mentioned task-specific frameworks on each of these tasks. We have repeatedly demonstrated these order-of-magnitude gains in efficiency and accuracy for PDE solving, learning and optimization, underscoring the utility of our paper. All of these are made possible by our key contribution: the novel FEM assembly algorithm that ensures a computational graph with O(1) nodes, unlocking downstream applications.
> > >
> > > We hope that this response addresses the reviewer's remaining concerns about our key contribution and kindly request them to upgrade their assessment accordingly.

---

### Decision · Program_Chairs · 2026-04-30

**Decision:**

Accept (regular)

**Comment:**

The work introduces a GPU-compatible Python-native framework, called TensorGalerkin,  for Galerkin-type finite element discretization. The framework is designed to provide a computational tool to accelerate and improve the accuracy of PDE solvers by addressing bottlenecks in traditional FEM and automatic differentiation–based operator learning through a Map-Reduce strategy. The framework targets efficient matrix assembly for time-dependent PDEs and operator learning tasks, combining Galerkin discretization with standard iterative solvers while optimizing operator representations via gradient-based methods. It features a two-stage kernel assembly process and supports multiple applications in a unified PyTorch pipeline. Experimental results on various PDE benchmarks demonstrate significant speedups over state-of-the-art neural operator learning approaches and existing Python-based solvers, while maintaining comparable accuracy, making it a strong engineering contribution with promising practical impact.

Three of the referees voted positively for the work, while another one voted negatively. The referees agree that the work has merit and is a nice contribution to the SciML field. Some concerns do remain, like novelty, the actual contribution in terms of engineering, and the impact of the work.

While I agree with some of the remaining concerns (novelty), I do acknowledge that such computational tools are often about the implementation and not necessarily algorithmic novelty. The authors do present very nice speedups over existing approaches, showing that the tool can be useful in this field. After weighing all pros and cons, I recommend acceptance of this work to this year’s ICML.